# Advancing lignin analytics via elucidation of linkage progressions in lignin populations
Filippa Ludvig [1,2,4], Åsa Emmer [3,4] & Martin Lawoko [1,2,4] ✉

The elucidation of the structures of individual lignin molecules in heterogeneous lignin isolates poses a challenge, underscoring the need for the development of robust analytical methods. Herein, we report the combined use of NMR and MALDI-TOF MS$^n$ as a facile approach for distinguishing and determining the detailed structures of individual oligomeric molecules in heterogeneous lignin mixtures. Supported by NMR, MALDI-TOF analysis of acetylated lignins provides precision by enabling the facile discernment of inter-unit linkages in lignin molecules. Furthermore, the progression of lignin linkages could be tracked through population studies, yielding a structural progression map that elucidates the chemical features of individual oligomers in milled wood lignins and synthetic lignins. By unmasking lignin's molecular heterogeneity, this study marks an essential milestone in lignin analytics with possibility to advance the frontiers of molecular-level research in both fundamental and applied lignin studies.

The complex structural motif of lignin arises from radical and cross-combinatorial coupling reactions in the cell wall's apoplastic domain[1–4]. Lignin polymerization occurs in proximity to the polysaccharides constituting the non-lignin components of the cell wall, resulting in a strong molecular association between these constituents. This intimate interaction renders the complete separation and extraction of unmodified lignin impossible[1,5,6]. Among the various mild extraction methods, milled wood lignin (MWL) is considered to retain a structure most closely related to the postulated native lignin structure present within the cell wall and middle lamella[5,7–9].

To characterize lignin structure, scientists employ an array of destructive and non-destructive analytical techniques. State-of-the-art methods, such as Fourier Transform Infrared (FTIR) Spectroscopy, Size Exclusion Chromatography (SEC), various types of Mass Spectrometry (MS), and Nuclear Magnetic Resonance (NMR) Spectroscopy, are among the available options[9,10]. Notably, 2D Heteronuclear Single Quantum Coherence (2D-HSQC) NMR Spectroscopy has emerged as the leading technique for analyzing inter-unit linkages in lignin. 2D-HSQC offers semi-quantitative insights that have enabled the construction of average structural motifs[2,11]. However, such averaging masks the structural heterogeneity of lignin populations and cannot provide insights into the exact structure of individual lignin molecules. There is a need to establish a methodological framework within lignin analytics that can distinguish populations and give the exact atomistic structure of intact individual oligomers.

SEC, used for determining lignin molecular weight, suffers from several limitations, including a lack of suitable lignin-like standards, strong analyte-column interactions that can alter the elution order, and dependence on the chosen calculation method[12–16]. In contrast, Matrix-Assisted Laser Desorption/Ionization Time-of-Flight (MALDI-TOF) MS holds potential to complement and extend NMR by facilitating studies of distinct structural populations and determining exact molecular weights.

MALDI-TOF is a soft-ionization MS technique that predominantly generates singly charged adduct ions $[M + X]^{+}$ with minimal chemical alteration[17]. In lignin research, this capability has enabled the analysis of both intact lignin and its degradation products—a field commonly referred to as lignomics. MALDI-TOF has proven valuable in identifying polymerization sequences, distinguishing between uncondensed and condensed aryl ether linkages, and studying Kraft, milled wood, and organosolv lignins[18–24]. Previous studies have noted the particular fine structure of MALDI-TOF MS spectra for synthetic, technical, and native-like lignins, though many of the cluster peaks remain unexplained[21–24].

[1]Wallenberg Wood Science Center, Department of Fibre and Polymer Technology, KTH Royal Institute of Technology, 100 44 Stockholm, Sweden. [2]Division of Wood Chemistry and Pulp Technology, Department of Fibre and Polymer Technology, School of Engineering Sciences in Chemistry, Biotechnology and Health, KTH Royal Institute of Technology, 100 44 Stockholm, Sweden. [3]Analytical Chemistry, Division of Applied Physical Chemistry, Department of Chemistry, School of Engineering Sciences in Chemistry, Biotechnology and Health, KTH Royal Institute of Technology, 100 44 Stockholm, Sweden. [4]These authors contributed equally: Filippa Ludvig, Åsa Emmer, Martin Lawoko. ✉e-mail: lawoko@kth.se

While MS techniques in lignomics provide molecular weight, structural alterations, and polymerization pattern data, tandem mass spectrometry (MS[n]) offers more profound insight into specific molecular structures[25–29]. MS[n] enables the selection and isolation of specific adduct ions from highly heterogeneous lignin mixtures, allowing detailed molecular-level analysis of individual species[26–29]. MS[n] systems can incorporate various ion sources and mass analyzers configured to detect either positive or negative adducts. Studies employing positive-mode tandem MALDI-TOF/TOF on lignin remain scarce. The most comprehensive to date, by Albishi et al., explored fragmentation pathways in MS[n] of lignin oligomers extracted from palm wood[27]. To our knowledge, no reported MALDI-TOF studies have exclusively examined laser-induced dissociation MALDI-TOF MS[n] on pure lignins. Furthermore, there is no prior art on targeted derivatization to distinguish inter-unit linkages and their progression in lignin oligomers by applying regular MALDI-TOF MS.

In this study, we propose the use of acetylation in combination with MALDI-TOF MS[n], guided by 2D-HSQC NMR, as an advanced approach to lignin structural analysis. This methodology enables the determination of the structural features of intact lignin oligomers present in MWL, as demonstrated herein on MWL isolated from Spruce. To provide a comprehensive overview of the identified oligomers in the sample, we propose "linkage progression maps" (LPMs) as a conceptual framework. In an LPM, the mass-over-charge of each identified adduct is given, as well as the inter-unit-linkage progression necessary to reach that exact mass. The LPM principle is based on the knowledge that lignin polymerization occurs

through radical coupling and primarily by end-wise addition of one monomer at a time[2,3]. Secondly, lignin polymerization occurs continually as long as the supply of monomers is available. Hence, at any given time, there will be oligomer populations that are differentiated by one monomer unit. Thus, if each linkage type can be made to contribute with a specific m/z increment when added to the growing oligomer, it becomes easy to follow the inter-unit linkage progression. We show herein that the fingerprinting of linkage type by m/z increment is achieved by acetylation of the sample before MALDI-TOF analysis. For full disclosure, there is an extensive amount of adduct peaks in the MALDI-TOF spectra of acetylated Spruce. Evaluating every single peak manually would be tedious work. We therefore showcase an LPM containing a series of oligomers derived from an identified starting adduct.

## Results and discussion

Despite several decades of studies, determining the exact molecular structure of isolated lignins remains an elusive analytical challenge. A detailed understanding of lignin's chemical properties is essential for comprehending plant cell wall biosynthesis, optimizing extraction processes, and designing lignin-based materials. In this study, we investigate the combined use of MALDI-TOF MS[n] and state-of-the-art NMR techniques to reveal distinct inter-unit linkage progressions within lignin structural populations. A general schematic of the mechanism is shown in Fig. 1, where the workflow traditionally used in lignin research to determine an average sample structure is contrasted with the methodology established in this

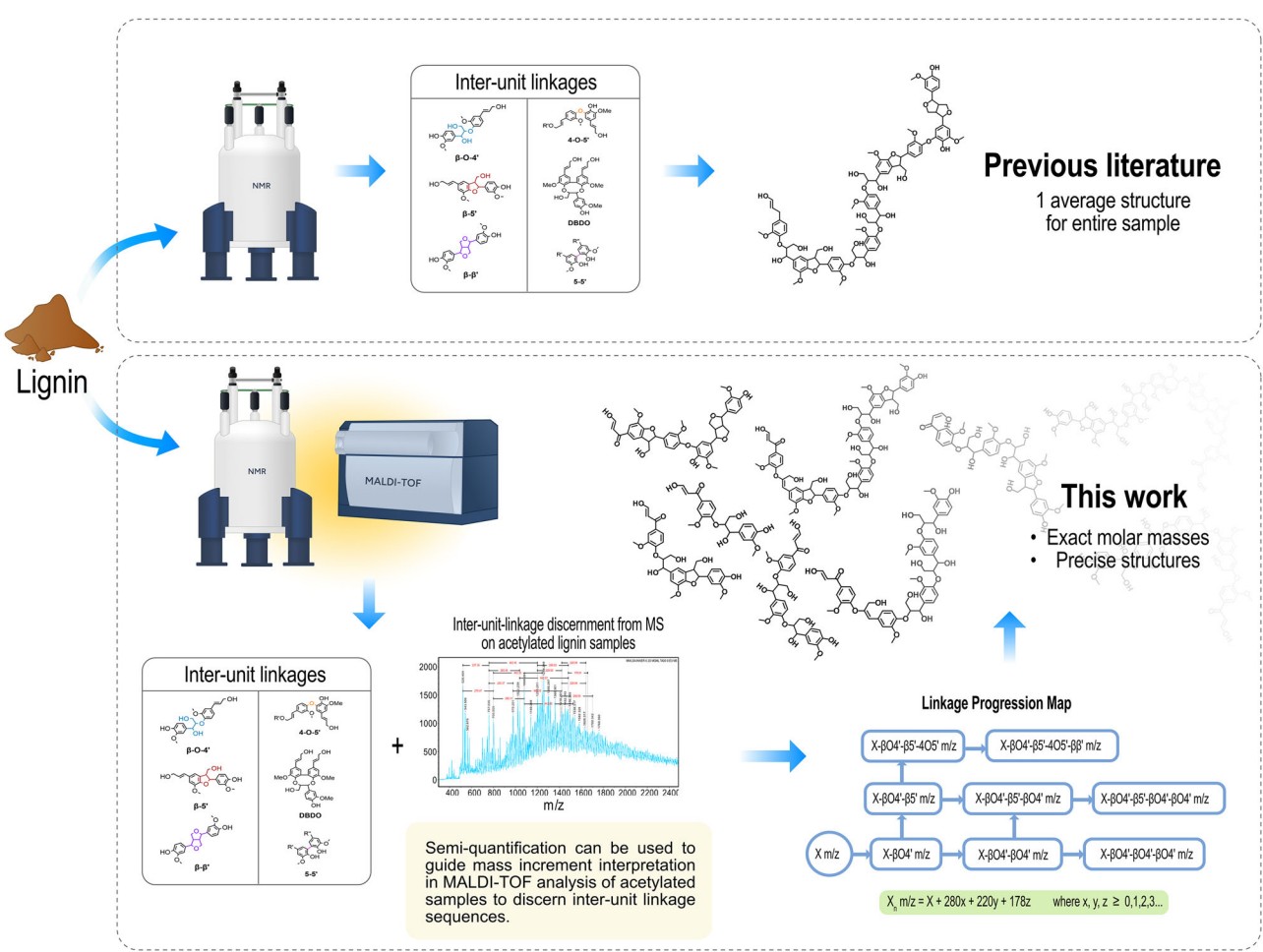

**Fig. 1 | Schematic for methodological framework developed within this project.** A comparison to already established lignin literature where 2D-HSQC NMR is used as a primary analytical technique to provide an average structural motif. While powerful, traditional 2D-HSQC NMR can only give an estimate of all inter-unit

linkages in a sample but not tie these to specific oligomeric structures. With the proposed workflow in this work, it is possible to reconstruct oligomers with their inter-unit linkage progressions tied to their exact molecular weights.

**Table 1 | Samples and relative content of bond type per total 100 aromatic units using 2D-HSQC NMR semi-quantification**

| Linkage | FEDHP | ENZDHP | MWLS |
|---|---|---|---|
| β-O-4' [%] | 18.9 | 14.1 | 36.9 |
| β-5' [%] | 13.6 | 25.9 | 13.7 |
| β-β' [%] | 12.3 | 13.1 | 6.0 |
| DBDO [%] | 1.2 | N/A | 3.1 |

The C2-H peak was used as internal reference, and all integrals were calculated per 100 aromatic units determined from C2-H region integral.

work. In addition to resolving lignin populations without prior fractionation, this work provides a framework for detailed structure determination of the individual molecules. This is achieved by utilizing complete acetylation of the lignin, which provides unique mass increments to each inter-unit linkage based on the differing number of hydroxyl groups in proximity to the linkage. Assuming end-wise polymerization, we can thus construct LPMs from the MALDI-TOF spectra to obtain the detailed structure of individual oligomers.

Synthetic lignins are not subjected to harsh chemical extractions that can alter the native molecular structures. These lignin templates serve as ideal reference substrates for the development of analysis. To support the analysis of Spruce MWL, we conducted additional studies on two types of synthetic oligomeric lignin mixtures. The identification and determination of relative content of inter-unit linkages was determined using 2D-HSQC NMR, while MALDI-TOF MS$^n$ was employed to investigate both molecular weights and linkage patterns.

## 2D-HSQC NMR analysis of synthetic and milled wood lignins

2D-HSQC NMR is a well-established technique for structural analysis of lignins, which was employed in this study to identify significant inter-unit linkages and assess differences in their relative abundance. Although this technique is not fully quantitative, it provides valuable semi-quantitative insights.

Enzymatic polymerization using horseradish peroxidase and $H_2O_2$ (ENZDHP) resulted in a more condensed synthetic lignin. A less condensed lignin, richer in β-aryl ether linkages, was synthesized through transition-metal catalysis using $FeCl_3$ (FEDHP) (Table 1). The $FeCl_3$ system had a pH of approximately 2, while the peroxidase system maintained a pH of roughly 7. These pH differences likely influenced the polymerization outcomes and highlight the impact of the polymerization environment on lignin structure. NMR spectra of MWL from Spruce (MWLS) were acquired using a 900 MHz instrument. The high resolution enables the detection of inter-unit linkages present at lower relative abundance. The MWLS 2D-HSQC NMR spectrum (Fig. 2A) displays common linkages such as β-O-4', β-β', and β-5', which are contrasted with those found in the synthetic lignins (Fig. 2B and C). As shown in Table 1, MWLS contains a significantly higher proportion of β-O-4' linkages. Additionally, weaker signal intensities tentatively assigned to α-O-4', α-O-alkyl', and γ-O-alkyl' linkages were observed (Fig. 2A)[30,31]. Signals corresponding to dibenzodioxocin (DBDO) were detected. However, there is ongoing debate as to whether DBDO functions as an actual branching point or merely as an end group[2,32]. In addition, signals from aldehydes are also observed (see the zoomed-in region in Fig. 2A). This specific region will be further discussed in relation to the MALDI-TOF MS$^n$ results.

In summary, the 2D-HSQC analysis provided a functional screening of the inter-units present in the lignin, which served as guidance for the MALDI-TOF MS inter-unit linkage progression analysis. All collected HSQC spectra can be found in Supplementary Figs. 13-15.

## MALDI-TOF MS analysis unravels linkage progressions in lignin

The MALDI-TOF MS spectra of synthetic lignins (Fig. 3) display distinct polymerization clusters with increments of 178 m/z and 196 m/z, as identified in the established literature. The increase of 178 m/z represents the

formation of a condensed bond such as β-β', β-5', and 4-O-5' in the end-wise polymerization of lignin by radical coupling. An increase of 196 m/z is a fingerprint for the formation of an alpha-hydroxylated β-O-4' bond. The increments are explained in Supplementary Fig. 12, and the spectra for pure 2,5-DHB, FEDHP, ENZDHP and MWLS can be found in Supplementary Figs. 1-4. As an example, the adduct 951 m/z is identified in both FEDHP and ENZDHP as a pentamer sodium adduct $[M + Na]^+$, comprised of 3 β-O-4' linkages and 2 β-5' linkages. Moreover, mass increments representative of the addition of condensed (178 m/z) and β-O-4' (196 m/z) inter-unit linkages can be measured between the most intense peaks in the entire spectra. The cluster analysis, therefore, unravels lignin polymerization events for the synthetic lignins. This polymerization occurs through well-established radical coupling, and, as the MALDI-TOF MS data suggest, mainly via end-wise addition of monolignols. At the same time, accurate molar masses (analyzed as m/z ratios) of lignin oligomers can be determined. Without further modification to the lignin samples, it is not possible to distinguish between condensed structures such as β-β', β-5', and 4-O-5' by MALDI-TOF MS, since all yield an increment of 178 m/z. The theoretical formation of 5-5' through only one monolignol addition would also produce a mass increment of 178 m/z, but this structure formation is the result of dimerization and not endwise addition[2]. Here, it is assumed that the observed mass increments of 178 m/z are mainly due to the formation of either β-β', β-5', or 4-O-5' since these structures were detected by 2D-HSQC NMR (Fig. 2 B, C). Interestingly, MWLS (Fig. 3, bottom) exhibited oligomer cluster patterns similar to those observed in the synthetic lignins. This suggests that a fraction of native lignin molecules remains unchanged during the production of MWL and has a low molecular weight. Alternatively, a selective cleavage of specified lignin bonds occurs during the preparation of MWL. This is further elaborated upon and discussed in the following sections.

## Acetylation followed by MALDI-TOF MS distinguishes condensed structures

To distinguish between condensed linkages in lignin that yield a mass increment of 178 m/z, acetylation was used as a chemical derivatization strategy. Acetyl groups are selectively introduced onto hydroxyl groups via esterification, allowing different linkage types to be distinguished based on their unique hydroxyl group count (Fig. 4).

For example, the β-β' (resinol) structure contains no aliphatic hydroxyl groups, whereas the β-5' (coumaran) structure contains one aliphatic hydroxyl group. Regardless, both sodium adduct of the dimers have a molar mass of 381 Da in their unmodified forms. While the sodium adduct ion of an unmodified β-5' dimer appears at 381 m/z, the corresponding acetylated sodium adduct appears at 507 m/z. This shift is due to the acetylation of one phenolic and two aliphatic hydroxyl groups. A 4-O-5' dimer, having the same number and type of hydroxyl groups, also yields an acetylated sodium adduct dimer mass of 507 m/z. Consequently, both β-5' and 4-O-5' structures result in a mass increment of 220 m/z in the acetylated cluster analysis and cannot be discerned from one another. In contrast, the sodium adduct of β-β' dimer shifts to 465 m/z upon acetylation due to the acetylation of two phenolic hydroxyl groups only, as no aliphatic hydroxyls are present. In the acetylated sample, β-β' formation still corresponds to a mass increment of 178 m/z, which enables the differentiation of β-β' from the β-5' and 4-O-5' linkages. The sodium adduct of β-O-4' dimer, initially observed at 399 m/z, appears at 567 m/z after acetylation. In a growing chain, the addition of three acetyl groups per monomer incorporated as a β-O-4' inter-unit moiety results in a mass increment of 280 m/z instead of 196 m/z. Depictions of sodium adduct dimer masses and their corresponding increments in both unmodified and acetylated forms are presented in Fig. 4.

Low-molecular-weight compounds are typically difficult to detect in MALDI-TOF MS due to overlap with the matrix signals. With lignin, acetylation effectively shifts adduct signals to higher m/z, which facilitates detectability. This effect was evident in the analysis of acetylated FEDHP, where the sodium adduct β-O-4' dimer became clearly visible

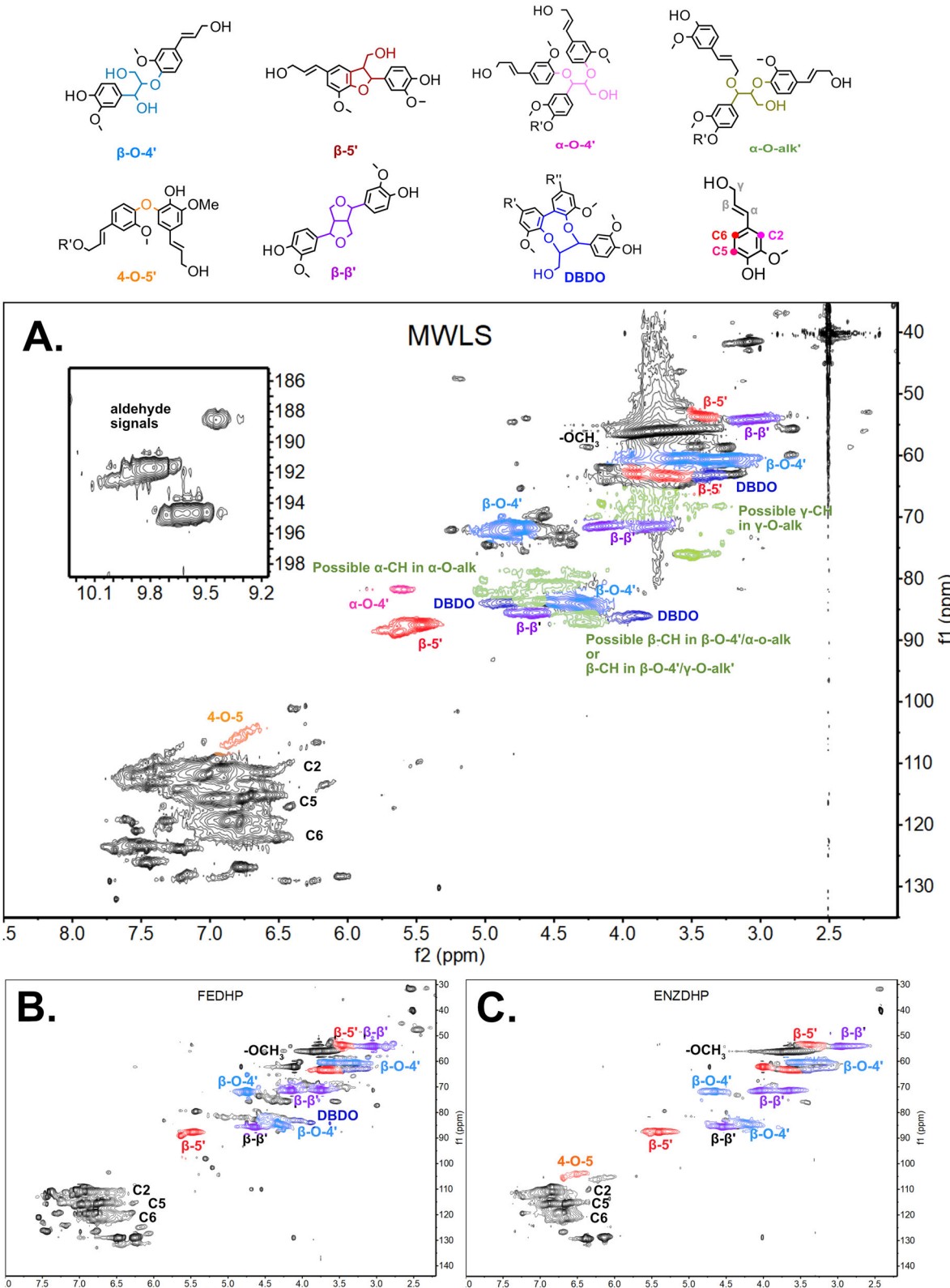

**Fig. 2 | 2D-HSQC NMR spectra of investigated MWL from Spruce, and synthetic lignins. A** Spectrum of MWLS acquired on a 900 MHz NMR instrument. **B** Spectrum of transition-metal catalysed DHP (FEDHP) acquired on 400 MHz NMR instrument. **C** Spectrum of enzyme-catalysed DHP (ENZDHP) acquired on 400 MHz NMR instrument. Identified inter-unit linkage types are marked in each spectrum.

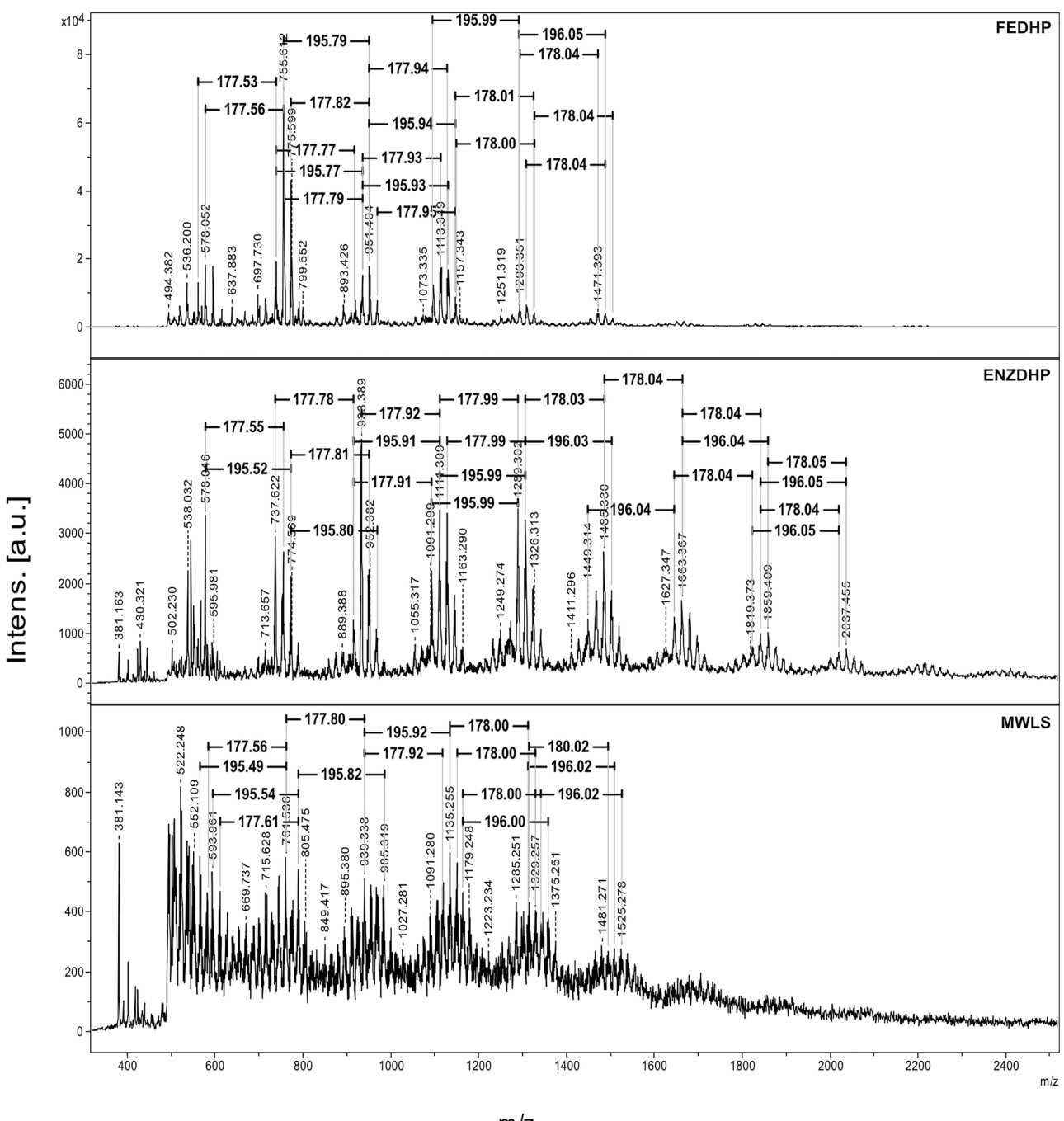

**Fig. 3 | MALDI-TOF MS spectra of unmodified lignins.** Top: FEDHP. Middle: ENZDHP. Bottom: MWLS. Mass increments between polymerization clusters are marked, showcasing primarily 178 m/z and 196 m/z increments representing the condensed and uncondensed bonds, respectively.

(Supplementary Fig. 5). The resulting mass spectra also showed shifted peak-to-peak increments of 280, 220, and 442 m/z, alongside some retained 178 m/z increments (Supplementary Fig. 5).

We evaluated the structure of MWLS after acetylation. $^{31}$P NMR (Supplementary Fig. 11) confirms complete acetylation of the sample. Two distinct crystallization sites were observed on the target plate (Supplementary Fig. 7) during MALDI-TOF MS analysis. The inner and outer crystallization regions were therefore sampled separately. The resulting spectra, as shown in Fig. 5, exhibited significant differences in absolute m/z values. This observation is due to a well-known segregation phenomenon associated with the 2,5-dihydroxybenzoic acid (2,5-DHB) matrix, commonly referred to as the "coffee-ring effect"[33]. However, both regions exhibited the expected mass increments for acetylated lignin between major cluster peaks.

Furthermore, several m/z values—such as 800, 1038, and 1140 m/z—were commonly observed in both regions, suggesting a consistent underlying oligomeric composition across the different crystallization domains.

Notably, the MALDI-TOF MS spectrum from the outer crystallization region (middle panel in Fig. 5) reveals several low m/z adducts that cannot originate from any of the unmodified dimeric structures typically involved in lignin polymerization—namely β–O–4′, β–5′, and β–β′ (residing at 567, 507, and 465 m/z, respectively). We propose that many of these modified dimeric structures result from homolytic cleavage reactions of lignin molecules occurring during the ball milling process, leading to the formation of Hibbert's ketone (HK)-like structures[34]. Supporting this hypothesis, the 2D-HSQC NMR spectrum of MWLS (Fig. 2A) indicates the presence of aldehyde-containing structures. The following section provides a more

**Fig. 4 | Dimer mass change and increment change for most common interunit linkages through acetylation.** 1a, β-5' (381 m/z)'. 2a, β-β' (381 m/z). 3a, β-O-4' (399 m/z). 4a, α-O-4'. 5a, 4-O-5'. 6a, 5-5'. 7a, DBDO. 1b, acetylated β-5' (507 m/z)'. 2b, acetylated β-β' (465 m/z). 3b, acetylated β-O-4' (567 m/z). 4b, acetylated α-O-4'. 5b, acetylated 4-O-5'. 6b, acetylated 5-5'. 7b, acetylated DBDO. *Suggestion for α-O-4' (or α-O-γ' as alternative) being an observed 442 m/z increment in acetylated sample. **5-5' is suggested by literature to occur as a dimerization reaction, and not as an addition of one single monomer to a growing chain. This increment is not included in the analysis, but the overall contribution of a theoretical monomeric unit to the observed 5-5' m/z is showcased to illustrate the mass contribution this unit has in an oligomer. A brief illustration of the acetylation mechanism employed for discernment is given at the bottom of the figure.

**Fig. 5 | Expansions of MALDI-TOF MS spectra of acetylated (MWLSA) and original (MWLS) Spruce MWL.** Top: Spectrum acquired from inner crystallization region of MWLSA. Middle: Spectrum acquired from outer crystallization region of MWLSA. Bottom: Spectrum acquired from original MWLS. Full spectra can be found in Supplementary Figs. 4, 6, and 7. Mass increments for 442 m/z at 758, 800, 978, and 1038 m/z is visible in inserted bottom spectrum for MWLSA.

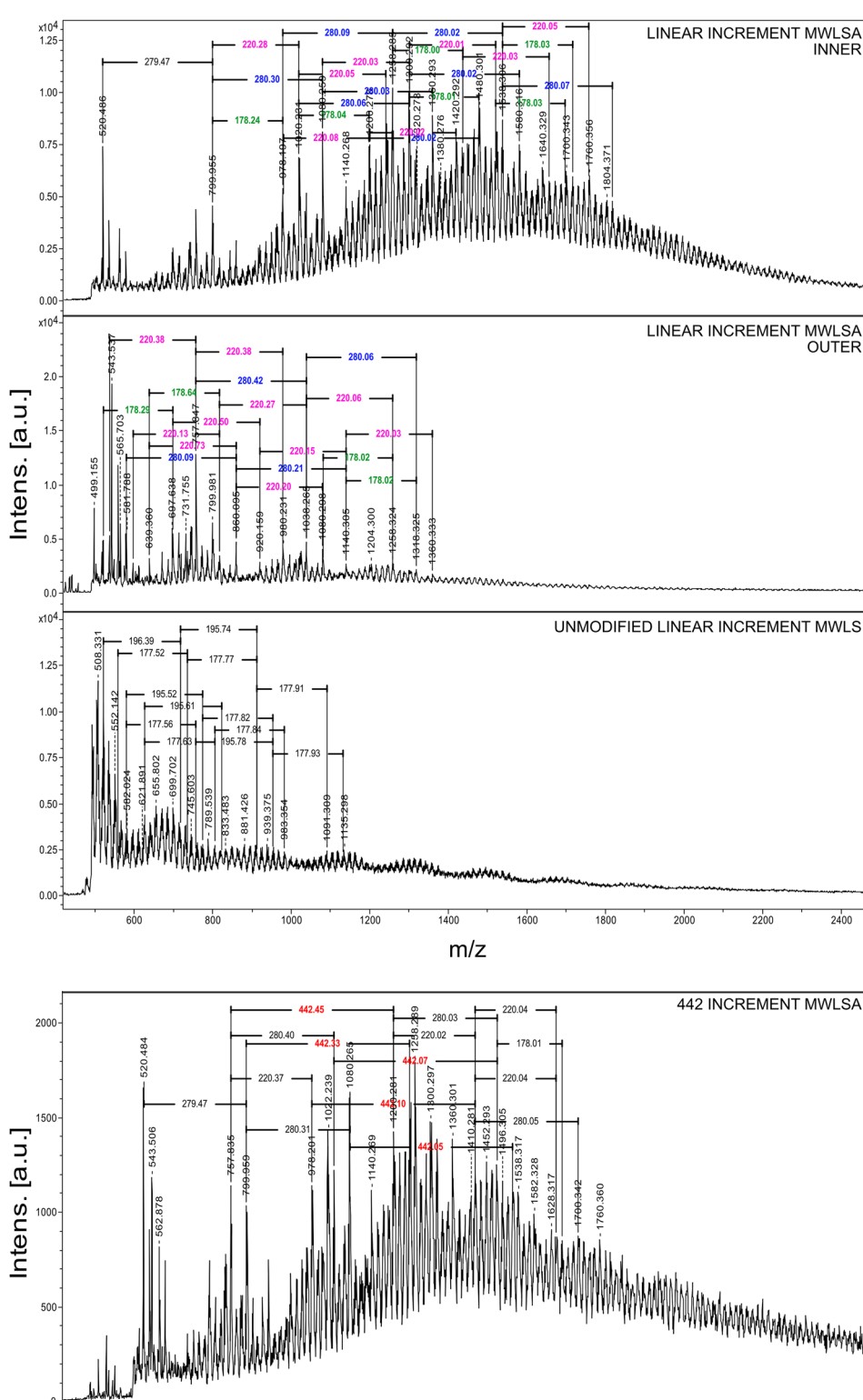

detailed discussion of the MALDI-TOF MS data for these unique lignin structures and their structural alterations.

### Unique m/z ratios unravel altered structure in spruce MWL

In MWLSA, three prominent adduct peaks—520, 758, and 800 m/z—were observed (Fig. 5, top). These adducts cannot be matched to known native

lignin structural motifs. We hypothesize that these reflect structural alterations resulting from the mechanical energy input during the MWL isolation process. Homolytic cleavage of β-aryl ether bonds, a well-documented phenomenon under high-energy conditions, may generate free phenolic ends and Hibbert's ketone-type degradation products[34,35]. We propose that these modified species likely give rise to the unique m/z values

**Fig. 6 | Suggested theoretical pathway during milling and/or MALDI-TOF events to produce sodium adduct 520 m/z in the acetylated sample.** The suggested delocalized electron is either present in MWL, or is the result of a MALDI-event.

observed in the MALDI-TOF MS spectrum, indicating either a partial breakdown or rearrangement of lignin's native oligomeric structure.

The tentative structure we proposed for the 520 m/z adduct (Fig. 6) exhibits signatures of homolytic bond cleavage and the presence of a stable radical. This radical appears to delocalize across the carbon framework within the same spin system, which may explain its survival. Observation of stable radicals in lignins has previously been made through electron spin resonance measurements[36–38]. The radical is also possibly generated during the MALDI process itself. Nevertheless, further investigations are required to validate the whole structure for the aliphatic end group. Regardless, the tentative structure is derived from a β–O–4′ moiety.

## Support for ketones and aldehydes derived by HMBC and MALDI-TOF MS²

Evidence for the altered structure was further investigated using HMBC (Fig. 7, full spectra in Supplementary Fig. 16), with the aromatic, allylic, and carbonyl regions of the combined HSQC and HMBC NMR spectra being specifically analyzed. There are clear correlations between aromatic protons ($^1$H: 7.3-7.5 ppm) with several carbons, including the carbonyl in α-ketone structures, in the HMBC spectra. Furthermore, correlations of aldehydic protons ($^1$H: 9.5-10.0 ppm) with allylic carbons are observed. A full description of the assignments is provided in the caption of Fig. 7. The lower aromatic rings on both structures were more difficult to verify by HMBC. The traditional Hibberts ketone, previously identified by NMR[34] was also verified in this work. The second ring containing the radical was not verifiable by the HMBC analysis. In general, the resolution of the oxygenated aliphatic region ($^1$H: 3.0-6.0 ppm in Fig. 7, HMBC spectrum) was very poor. In this region, only correlations related to β-5, α in Hibberts ketone, and methoxyls were observed. Some overlap of signals occurs, for example, between methoxyl signals and those pertaining to ketones (Fig. 7A). Despite overlap issues, HMBC provided further support for the tentative structure.

MS² further elucidated the nature of these altered structures. MALDI-LIFT-TOF/TOF parent spectra of the selected adducts revealed evidence of metastable fragment ions, which arise from post-source decay in the field-free drift region (Fig. 8). These fragment ions maintain the same velocity as their precursor ion and can therefore only be detected via MS² analysis. The MALDI instrument employs a proprietary LIFT technique, which reaccelerates ions by imparting kinetic energy, thereby enabling the separation of fragment ions from their precursor ions[39]. When parent spectra were collected, partial fragmentation occurred for the analyzed adducts when using α-Cyano-4-hydroxycinnamic acid (HCCA) as a matrix. This is visible in the MS² parent spectra for the adducts at m/z 1039 and 1081 (Fig. 8). When laser intensity was increased to acquire the respective adduct fragment spectra, these peaks remained the most intense (Fig. 8). Additional low-intensity fragment ions also emerged, confirming further fragmentation under high-energy conditions (Fig. 8).

In MS², HCCA was used in place of 2,5-DHB as a matrix since HCCA typically promotes more extensive analyte fragmentation. Attempts to

collect meaningful MS² spectra using 2,5-DHB were unsuccessful. With HCCA, a slight shift in recorded mass was sufficient to cause an apparent +1 Da increase in m/z compared to the masses obtained with 2,5-DHB. In MS², the parent ion selected for fragmentation was the peak closest to the expected m/z value observed in regular MS mode, which further explains the slight deviation in m/z between precursor ions observed in MS and MS² spectra caused by the rounding effect.

Low m/z fragments consistently appear across the investigated adduct ions (Fig. 8, Supplementary Figs. 8 and 9 for sample spectra, and Supplementary Fig. 10 for HCCA matrix spectra). The most frequently observed fragment ions occur at 80, 122, 152, 179, 221, 258, 480, and 522 m/z. After 522 m/z, fragmentation seems to be more dependent on which specific possible cleavage sites are available within the chain for the investigated adduct.

Using the sodium adduct ion at m/z 1039 as a representative example for MS² analysis on acetylated lignins we propose a possible fragmentation mechanism (Fig. 9), assuming that this species is a linear oligomer with or without a pendant group. Starting from the parent ion, fragmentation proceeds via loss of two hydrogen atoms, followed by a sequence of homolytic bond cleavages and stepwise deacetylation reactions. Notably, the mass increment between 522 and 802 is 280 m/z, which may signify a β-O-4' unit in the middle of the chain. This series of events ultimately yields several low m/z fragments, including the highly abundant ion at m/z 258, which is structurally consistent with the proposed alteration, further aligning with the HMBC result.

Possible fragmentation pathways of condensed linkages were investigated for the adducts but no plausible fragment masses corresponding to the observed MS² peaks could be reconstructed. In summary, the MS² analysis further supports the proposed structural alteration in lignin, originating from a β-O-4' subunit. However, further validation is required and could be achieved through selective chemical fragmentation. This would provide candidate structures and enable alternative identification of pathways.

## Linkage progression in Spruce MWL oligomer populations

After unravelling the aliphatic end group structure, the linkage progression from the low m/z is constructed in accordance with the earlier described LPM principle. More specifically, using the 520 m/z (Fig. 5, top spectrum) adduct peak as a molecular anchor, the progression of lignin inter-unit linkages can be studied based on the previously discussed mass increments observed after acetylation. A comprehensive analysis is summarized in Fig. 10, covering structural populations ranging from dimers (520 m/z) to heptamers (1818 m/z) in an LPM. For this simplified analysis, a mathematical model is constructed to explain the found adducts as follows: $520 + 280x + 220y + 178z$ (where x, y, z ≥ 0, 1, 2, 3,...). To reiterate, the 280 m/z signifies the β-O-4' linkage, 220 m/z is either 4-O-5' or β-5, and 178 m/z is the β-β' linkage. Note here that if there is a β-β' increment, the 220 residing prior in the LPM must be a 4-O-5' due to the directionality of the chain. From the LPM (Fig. 10), we categorize the oligomer populations into

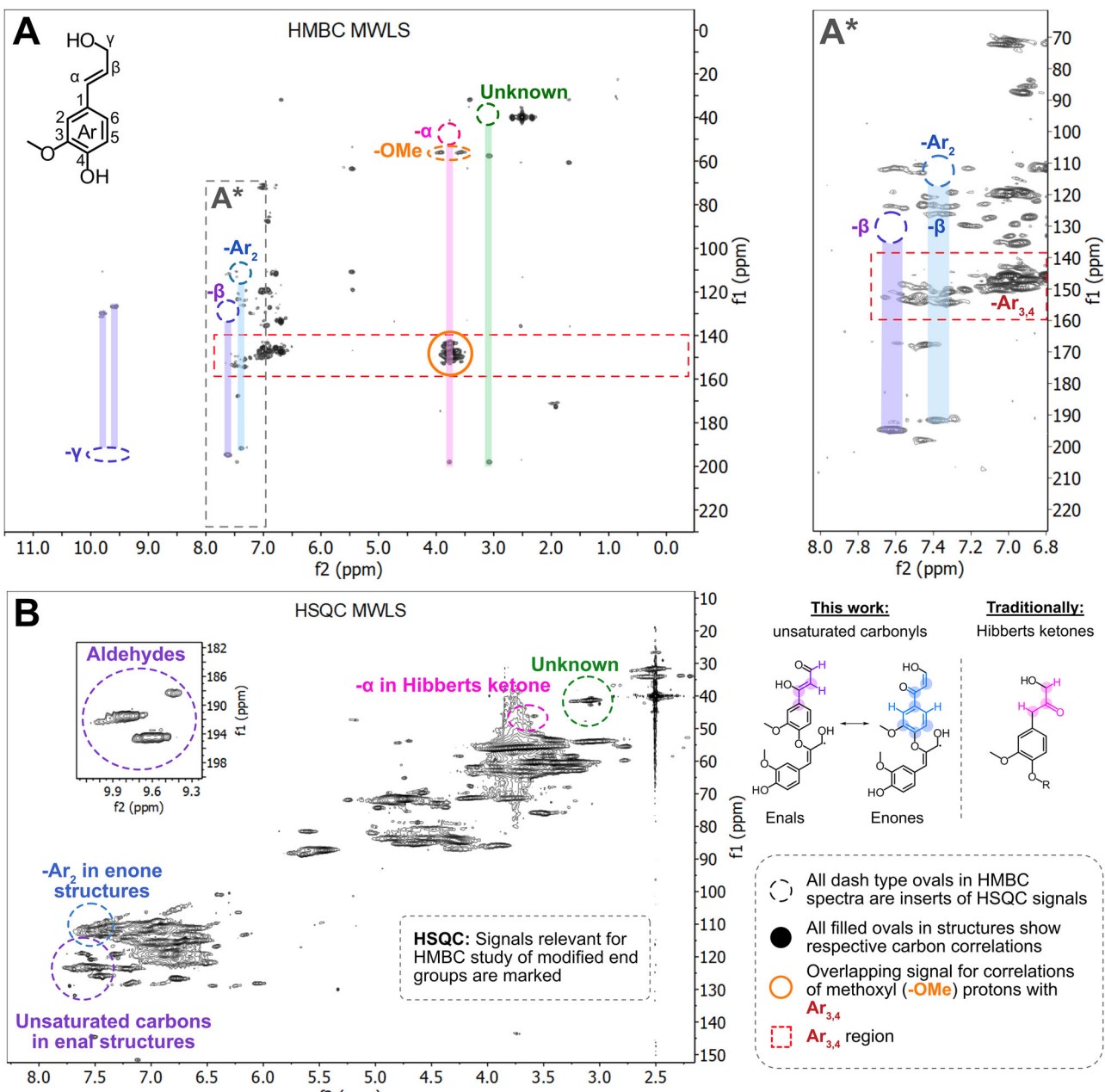

**Fig. 7 | Deciphering the atom connectivity in altered structures through combined HMBC (400 MHz) and HSQC (900 Mhz) NMR.** Top left (**A**): Complete HMBC NMR demonstrating atom connectivity by through bond correlations. Purple coded: the aldehyde proton connectivity to $C_\beta$, $C_\alpha$ in allylic structures, typically appearing in the region $^{13}C$: 125.0-130.0 ppm, and $C_1$ aromatic typically around $^{13}C$: 128.0-130.0 ppm. Blue coded: the aromatic proton attached to $C_2$ correlates with aromatic carbons $C_1$, typically at $^{13}C$: 128.0 ppm, $C_3$ and $C_4$, typically at $^{13}C$: 145.0-152.0 ppm, and $C_\alpha$ in ketone, typically at $^{13}C$: 190.0-210.0 ppm. Pink

coded: previously analyzed Hibberts ketone as reported from model studies by Miles-Barett et al. [34] is also detected. Top Right (**A***): Zoom in of the aromatic, allylic and aldehyde regions of the HMBC. Signals of specific interest are marked and color-coded. Bottom left (**B**): 2D-HSQC NMR with noted signals relevant for HMBC analysis. Bottom right: Classification of altered structures detected in this work. Description of used symbols and previously analyzed ketones, also detected in this work.

three groups: (1) Linear oligomers composed of block-like co-oligomers, where segments containing exclusively β-O-4' bonds are followed by segments dominated by condensed structures (Fig. 10A). (2) Structurally homogeneous oligomers composed solely of aryl ether-bonded lignin monomers (Fig. 10B). (3) Oligomers with inter-unit linkages exhibiting no specific sequential order, not shown in the figure.

For all m/z values included in the LPM, a simple statistical analysis has been carried out. Mean values, standard deviations, and relative standard deviations have been calculated for S/N and m/z and are provided (Supplementary Tables 1 and 2) for each crystallization region, respectively.

## Ambiguous m/z increment of 442

A mass increment of 442 m/z, observed from the 758 m/z adduct (Fig. 5, inserted spectrum at bottom) may suggest the presence of etherified pendant groups in lignin. This linkage may be explained by nucleophilic addition reactions to quinone methide intermediates formed either during lignin polymerization or during lignin aging[30]. The hypothesized structure is illustrated (Figs. 4, 4B). Unlike the inter-unit linkages discussed earlier, these structures—primarily α-O-4' and α-O-γ'—remain controversial due to the lack of unequivocal NMR evidence confirming their presence. However, some progress has been made in analyzing these structures and their origins.

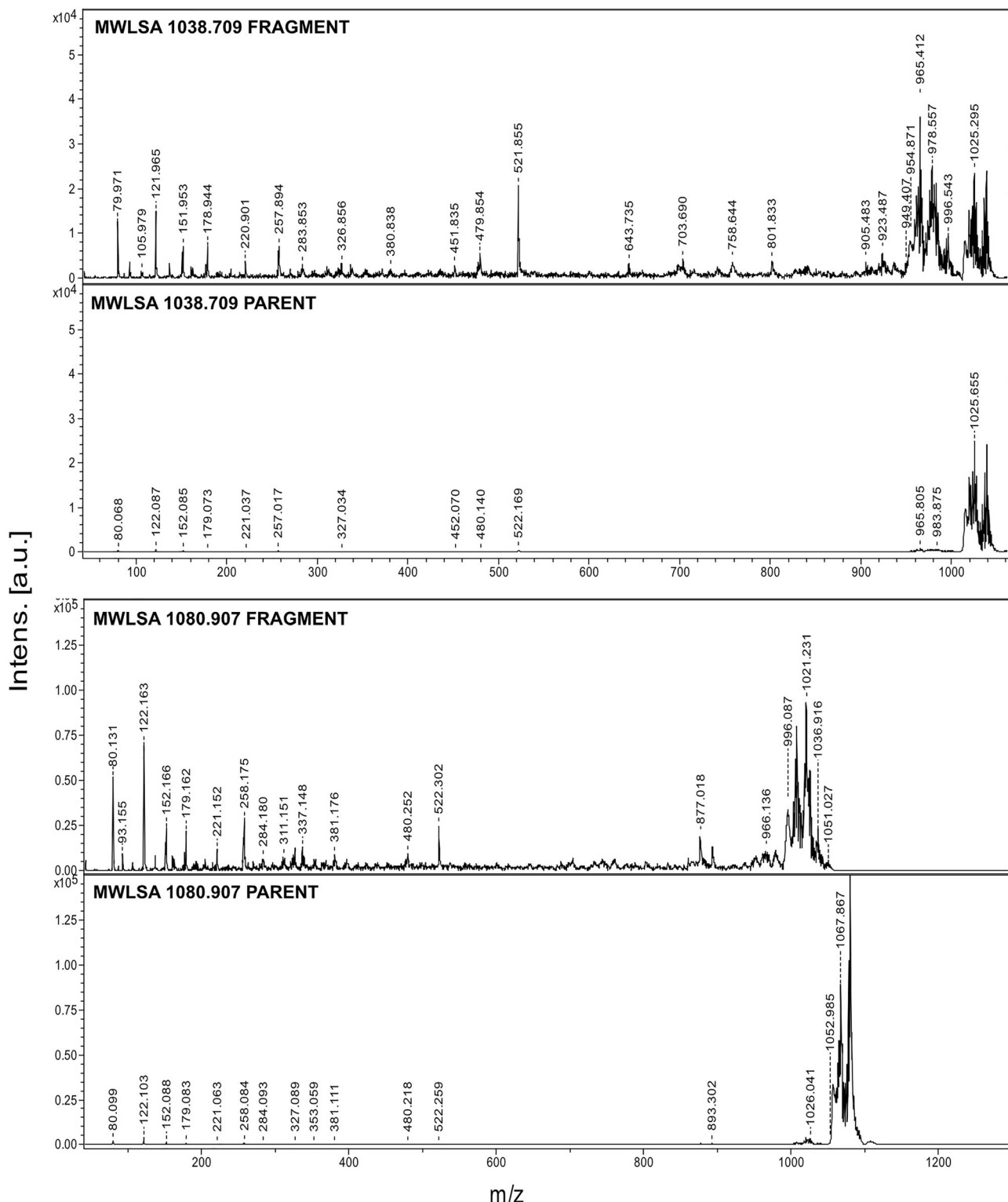

**Fig. 8 | MS² parent and fragment spectra of lignin oligomers in MWLSA obtained by MALDI-LIFT-TOF/TOF.** MALDI-LIFT-TOF/TOF spectra of adducts 1039 (top) and 1081 (bottom) m/z. Parent spectra is visible in bottom, and fragment spectra is visible in top for each sample. Fragment peak 522 m/z is found in both samples, along with other common low m/z fragment peaks. Notice the 280 m/z distance between 522 and 802 m/z in 1039 fragment spectra, signifying a β-O-4' unit.

Zhu et al. performed model studies and successfully matched the 2D-HSQC resonances of the models with the suspect resonances of these structures observed in MWL[30]. The next step forward would be determining the atom connectivity using techniques such as HMBC and HSQC-Total Correlation Spectroscopy (TOCSY). The HMBC study in this work (Fig. 7) was unable to provide any evidence due to low concentrations of these linkages, making the direct linkage analysis of the bonds almost impossible without enrichment of the linkages by other methods. Therefore, further work is needed to provide unequivocal evidence of their existence. One approach would be to enhance the signals related to these inter-units by fractional enrichment, then subject them to HSQC, HMBC and HSQC-TOCSY analysis. This is a subject for future work.

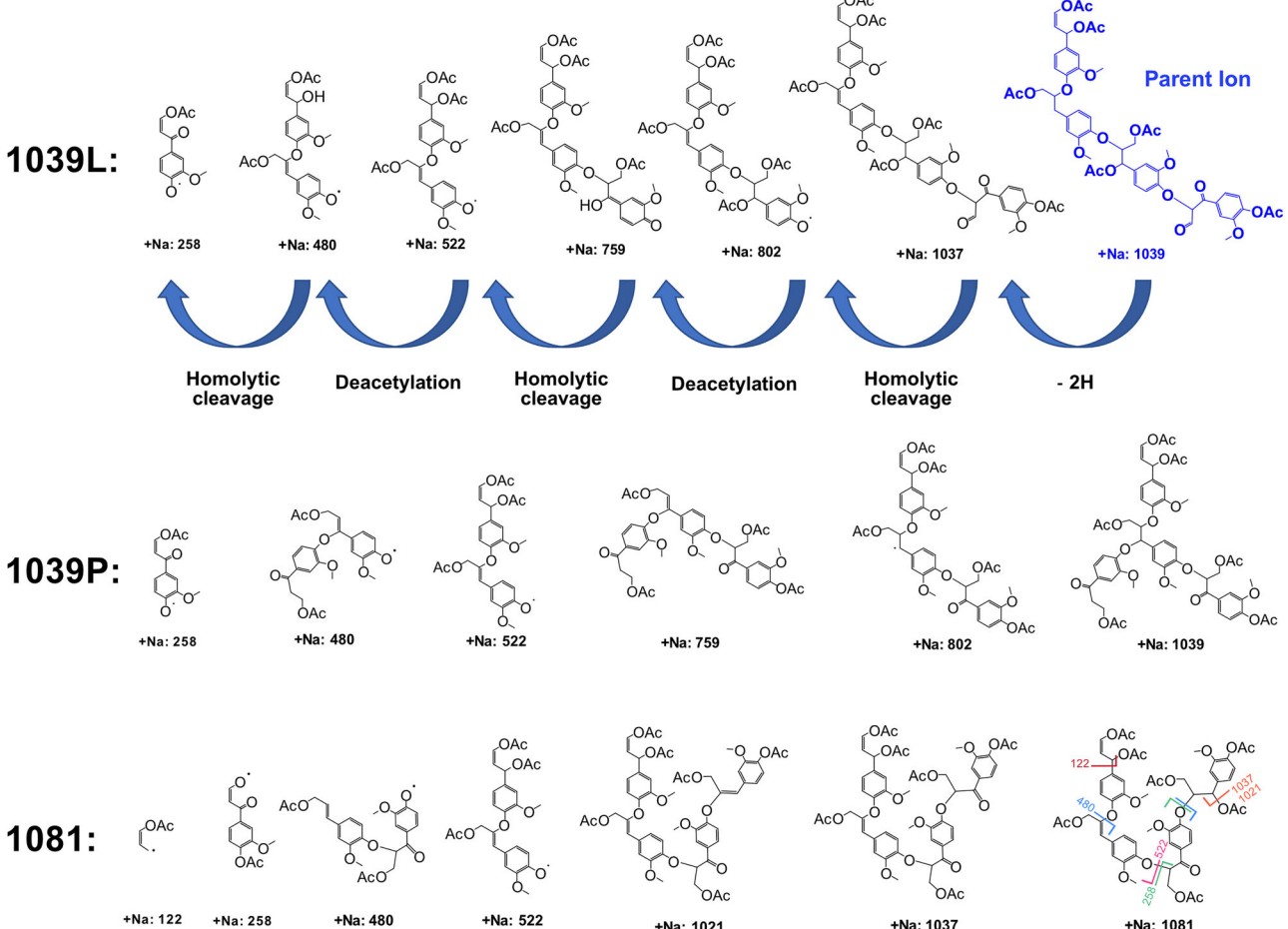

**Fig. 9 | Reconstruction of adducts investigated with MALDI-LIFT-TOF/TOF MS².** Top: suggested fragmentation mechanism of assumed linear acetylated β-O-4' lignin oligomer (1039 L) in MS². Middle: suggested reconstructed fragment for a possible structure containing a pendant group (1039 P). Bottom: suggested reconstructed linear 1081 structure.

## Limitations and applicability

Certain limitations of MALDI-TOF MSⁿ should be taken into consideration. Most notably, there is discrimination against higher molecular weight oligomers in heterogeneous samples, resulting in a skewed molecular weight distribution with a lower average[40,41]. Matrix compatibility, sample preparation, and sufficient crystallization are crucial steps for MALDI-TOF MSⁿ analysis, significantly influencing the recorded spectra[42]. It should be duly noted that in this publication, the work exclusively focuses on the low-molecular-weight oligomers present in MWL from Spruce. Specific lignin oligomers containing 5-5'/DBDO inter-unit linkage types are the product of dimerization reactions and will not be easily discerned with the currently developed method of analysis. The established framework is primarily suited for oligomers resulting from end-wise polymerization, as the increments between polymerization clusters are considered. Moreover, MALDI-TOF MSⁿ is predominantly qualitative, rather than quantitative. While the tentative structures of low-molecular-weight oligomers residing in the sample are identified, it is not possible to determine their relative ratio. Therefore, semi-quantitative 2D-HSQC NMR remains necessary for quantifying inter-unit linkages.

Concerning applicability, the general concept on which this methodology is based must be considered. In its essence, the acetylation reveals the number of hydroxyl groups present in each lignin inter-unit linkage, making it possible to discern molecular mass differences via MS before and after derivatization. This difference is the principle that allows for the determination of structural unit and inter-unit linkage progression. However, further validity across various types of technical and native-like lignins is required.

We have conducted a preliminary analysis of softwood Kraft lignin in the unacetylated and acetylated forms (Supplementary Figs. 17-19). When compared to the acetylated MWL from spruce, there is no apparent clustering in the spectra that reflects the presence of native inter-units (i.e. the 280, 220, and 178 m/z increments). However, between some of the most intense peaks in the acetylated Kraft sample, a recurring increment of 88 or 176 can be found, but we have not been able to postulate a structure for this observation. Crude technical lignins are known to be highly heterogeneous with structural features which are still poorly understood. Further studies are warranted and may involve fractionation before analysis.

## Conclusion

State-of-the-art NMR spectroscopy has played a crucial role in lignin structural studies over the past three decades. Nevertheless, the heterogeneous nature of lignin complicates the structural determination of individual molecules residing within these complex mixtures. Consequently, the field has traditionally relied on an average structural motif derived from knowledge of inter-unit linkages to describe a global lignin architecture.

In this study, we demonstrate that combining NMR with MALDI-TOF MS provides sufficient resolution to analyze lignin inter-unit linkage progressions in acetylated lignin samples. Acetylation introduces an additional layer of specificity, enabling the discrimination and discernment between the most common condensed bonds. This framework was successfully applied to Spruce MWL and revealed the linkage progressions originating from a selected low-molecular-weight dimer. The resulting LPM elucidates the structural features of lignin oligomers up to seven units long, thereby revealing distinct populations within the low-molecular-weight portion of

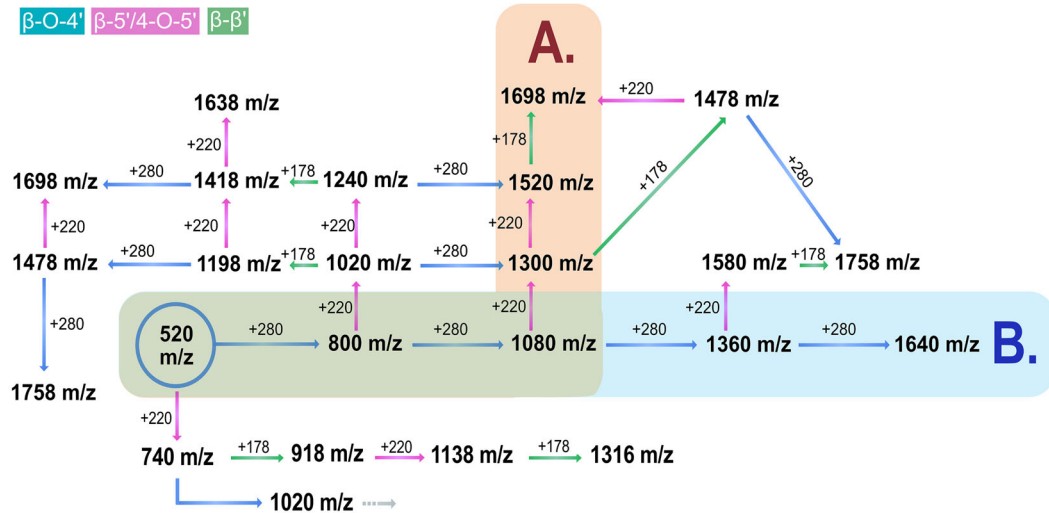

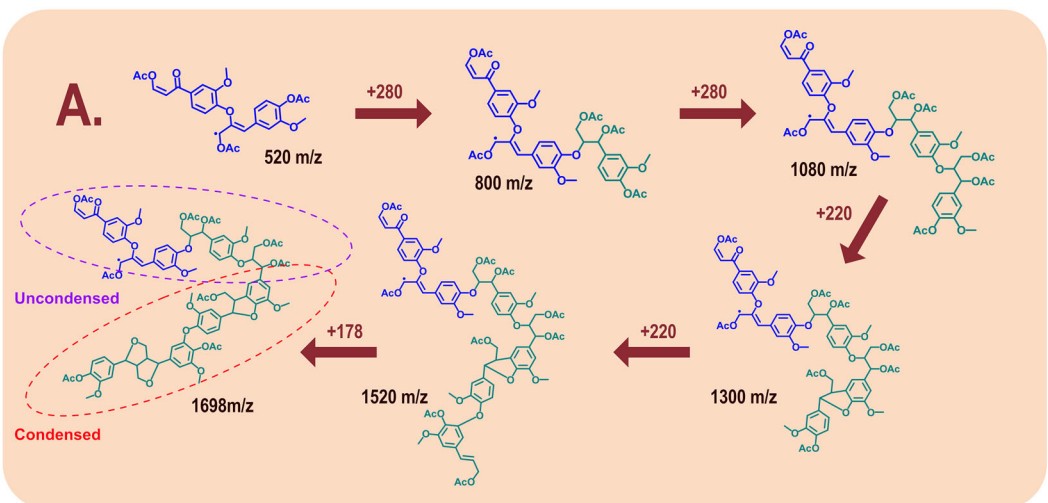

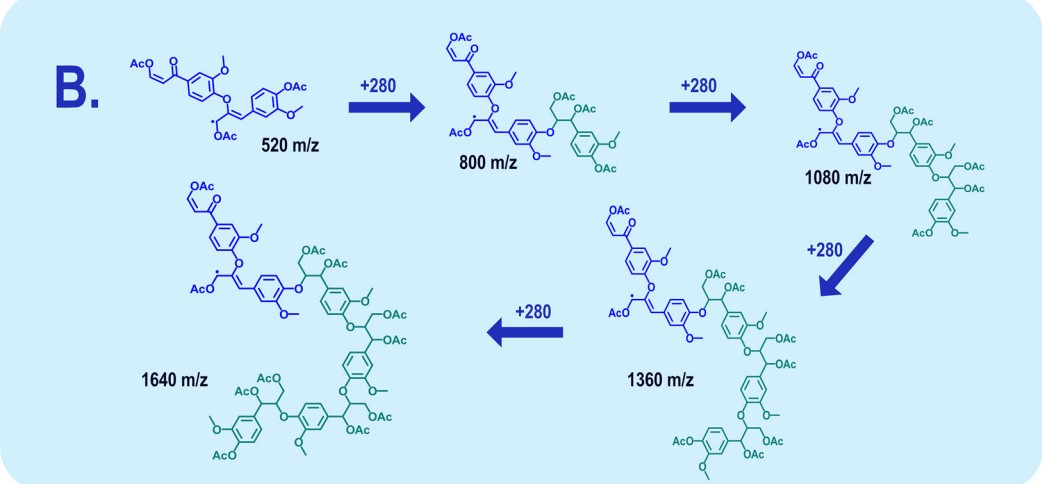

**Fig. 10 | Linkage progressions in lignin populations (LILIPOPS) illustrated in an LPM.** Constructed inter-unit linkage progression maps from MALDI-TOF MS data on MWLSA from 520 m/z adduct ion for linear oligomers. Two examples of linkage patterns showcasing block oligomers (**A**) and homo-oligomers (**B**) are included. Unordered oligomers are found in the LPM, but not showcased in the figure. All m/z include Na$^+$ adduct.

the sample. The identified linear oligomers were further categorized into homo-oligomers, block oligomers, and oligomers exhibiting irregular inter-unit ordering. Overall, this work advances our understanding of lignin heterogeneity and represents a significant step forward in lignin analytics. Further validation of the methodology across different lignin types is required to define the application space.

## Methods
### Materials
$N,N$-Dimethylformamide (DMF) (anhydrous) 99.8%, Coniferyl aldehyde 98%, Iron(III) chloride hexahydrate reagent grade ≥98%, Hydrogen peroxide solution 30 wt% in $H_2O$, Dichloromethane, $MgSO_4$ (anhydrous) ≥99.5%, $NaBH_4$ ≥ 98%, pyridine (anhydrous) 99.8%, $CDCl_3$-d ≥ 99.8 at. % D, endo—N-hydroxy-5-norbornene-2,3-didicarboximide (eHNDI) 97%, Chromium(III) acetylacetonate (Cr(AcAC)$_3$) 99.99%, 2-chloro-4,4,5,5-tet-ramethyl-1,3,2-dioxaphospholane (CI-TMDP) 95%, Acetic anhydride ≥99%, NaCl ≥99%, 2,5-dyhydroxybenzoic acid (DHB) for MALDI-MS ≥ 99%, Trifluoroacetic acid 99%, and Toluene ≥99.5% was obtained from Sigma-Aldrich. Peroxidase from horseradish type VI (essentially salt free, lyophilized powder) was obtained from Sigma Life Science. Acetonitrile ≥99.9% was obtained from Honeywell Riedel-de Haën. Acetone ≥99.5%, and MeOH 99.8% was obtained from VWR Chemicals. Ethyl acetate was obtained from Supelco. MilliQ was obtained from Millipore Ultrapure System equipped with MilliPak 0.22 μm filter. α-cyano-4-hydroxy-cin-namic acid (HCCA) for MALDI-TOF MS was obtained from BRUKER Daltonik GmbH. Milled wood lignin from Spruce, prepared by the Bjork-man procedure[8], was a kind gift from the laboratory of Dr. Nilvebrant at Borregaard.

### Reduction of coniferyl aldehyde
Coniferyl aldehyde (3.030 g) was added to ethyl acetate (180 mL) under stirring. $NaBH_4$ (1.401 g) was added to the mixture under stirring. The reaction flask was covered in aluminum foil and left to stir overnight. The reaction was quenched using MilliQ (200 ml), and was left for 30 min. After quenching, the reaction mixture was transferred to a separatory funnel. The aqueous phase was extracted with ethyl acetate. The organic aliquots were pooled together and washed with brine, followed by drying with $MgSO_4$. The slurry was vacuum filtrated over a glass fiber filter (Whatman) and the collected organic phase was rotary evaporated. The product was re-dissolved in boiling DCM and transferred to a glass jar, which was scraped with a glass rod in the bottom to induce crystallization. The product was left overnight in a freezer. The crystallized product was filtered and dried to remove any residual solvent. $^1H$ NMR confirmed complete conversion to coniferyl alcohol (CA).

### HRP synthesis of synthetic lignin
HRP (3 mg) was dissolved in MilliQ (20 mL) and left under gentle stirring (150 RPM). Aqueous $H_2O_2$ (0.1 M, 20 mL) was prepared. CA (360 mg) was dissolved in acetone (1.29 mL) and diluted with MilliQ (18.71 mL). The substrate solutions were injected into the enzyme solution using a NE-1800 eight channel programmable syringe pump at a constant rate (750 μL h$^{-1}$, 150 RPM, RT, 48 h). After injection, the reaction was allowed to stir additionally (RT, 150 RPM, 24 h). The suspension was centrifuged using a Rotina 420 (2000 RPM, 30 min) and washed once with MilliQ. The supernatant was decanted and the pellet was lyophilized to yield ENZDHP.

### FeCl$_3$ synthesis of synthetic lignin
CA (1.032 g) was dissolved in acetone (21 mL) and added to MilliQ (205 mL) in a round-bottom flask while stirring. FeCl$_3$ * 6 $H_2O$ (1.702 g) was diluted in MilliQ (6 mL) and added to the coniferyl alcohol solution. The round-bottom flask was covered in aluminum foil and left to stir (24 h, RT). The reaction mixture was transferred to a separatory funnel and extracted with ethyl acetate. Organic phase aliquots were pooled and washed with brine followed by drying with $MgSO_4$. The slurry was vacuum filtrated over a glass fiber filter (Whatman) and the collected liquid phase was rotary

evaporated. The dried product was extracted from the flask using acetone, and dried overnight. This yielded FEDHP.

### Acetylation of lignins
Acetylation was carried out according to previously established protocol[43]. Round-bottom flasks and glass corks were dried in an oven (105 °C, 5.5 h). Acetic anhydride and pyridine were mixed (1:1 v/v, 2 mL) and added to a dried round-bottom flask. Approximately 100 mg of lignin was added to the reaction liquor under stirring. The reaction round-bottom flask was covered in aluminum foil and left stirring overnight. The reaction was quenched with MeOH (25 mL) for 30 minutes in an ice bath. Toluene (3 ×10 mL) was added to remove the pyridine, followed by rotary evaporation to yield the acetylated lignins.

### $^{31}$P NMR
A Bruker Avance III HD 400 MHz (Bruker Corporation, Billerica, MA, USA) instrument, BBFO probe equipped with a Z-gradient coil (5 mm PABBO BB/19F-1H/D Z-GRD Z116098/0174) was used for $^{31}$P NMR.

$^{31}$P NMR was conducted according to previously established protocol[44,45]. Internal standard (eHNDI) (30 mg) was added to pyridine (500 μL). Cr(AcAc)$_3$ (approximately 3 mg) was added to the eHNDI solution. Lignin (30 mg) was weighed. DMF (100 μL) and pyridine (100 μL) was added to the lignin and was vortexed until dissolution (30 °C, 30 min). eHNDI solution was added (50 μL) and the sample was vortexed (30 min). 2-chloro-4,4,5,5-tetramethyl-1,3,2-dioxaphospholane (100 μL) and CDCl$_3$ (450 μL) was added drop-wise to the mixture. The reaction was allowed to proceed for 30 min before the solution was transferred with a Pasteur pipette to an NMR tube.

NMR method "N P31ig (P31 for lignin)" was used. $^{31}$P NMR. Number of scans: 256. 5 s relaxation delay.

Spectrum was analyzed with MestReNova software.

### 2D NMR
A Bruker Avance III HD 400 MHz (Bruker Corporation, Billerica, MA, USA) instrument, a 5 mm BBFO probe equipped with a Z-gradient coil (5 mm PABBO BB/19F-1H/D Z-GRD Z116098/0174) was used for structural analysis of synthetic lignins.

For synthetic lignin, 2D-HSQC spectra were obtained with 'HSQCETGPSI'. 88 scans over 512 × 256 increments with 1.4275 s relaxation delay and 0.1710 s acquisition time, performed at 298.0 K. Spectra were analyzed with MestReNova software (ver. 14.2.0, Mestrelab Research). Data processing included Fourier-transformation, and phase and baseline correction were carried out in both dimensions using a third-order Bernstein polynomial fit. The C2-H region of the aromatic ring was used as an internal standard.

A 900 MHz Oxford magnet equipped with Bruker Avance HDIII console (Bruker Corporation, Billerica, MA, USA) and 3 mm TCI 1H/13 C/ 15 N cold probe (CP TCI 900SA H-C/ N-D-03 Z) was used for 2D-HSQC structural analysis of milled wood lignin from Spruce (MWLS).

For MWL, HSQC spectra were obtained with 'HSQCE-DETGPSISP2.4'. 24 scans over 1153 × 512 increments with 1.0000 s relaxation delay and 0.0799 s acquisition time, performed at 298.0 K. Spectra were analyzed with MestReNova software (ver. 14.2.0, Mestrelab Research). Data processing included Fourier-transformation, and phase and baseline correction were carried out in both dimensions using a third-order Bernstein polynomial fit. The C2-H region of the aromatic ring was used as an internal standard. When edited HSQC is used for semi-quantification, the integral signals must be adjusted by their DEPT signal intensity at 135°. CH: sin(135°), CH2: sin(2 × 135°), and CH3: 3×sin(135°)cos(2 × 135°).

A Bruker 400 DMX (Bruker Corporation, Billerica, MA, USA) instrument equipped with a 5 mm probe (PA BBO 400S1 BBF-H-D-05 Z) at room temperature was used for HMBC structural analysis of milled wood lignin from Spruce (MWLS).

For MWL, HMBC spectra were obtained with 'hmbcgpl2ndqf'. 200 scans over 512 × 256 increments, 1.5000 s relaxation delay and 0.1065 s

acquisition time, performed at 298.0 K. Spectra were analyzed with MestReNova software (ver. 14.2.0, Mestrelab Research). Data processing included Fourier-transformation, and phase and baseline correction were carried out in both dimensions using a third-order Bernstein polynomial fit.

80 mg of sample was dissolved in 800 μL of DMSO-$d_6$ in a glass vial with Teflon cap followed by vortex until complete dissolution. It was possible to dissolve all samples at RT without addition of heat, and samples were dissolved readily after addition of solvent and light vortex.

## MALDI-TOF (MS)

Spectra were acquired on a BRUKER ultrafleXtreme (Bruker, Bremen, Germany) using an Nd:YAG laser (355 nm) in reflectron mode. For each spectrum, 4000 laser shots were used for the generation of spectra in the 0-5000 m/z region in positive mode. The data was acquired using flexControl 3.4 processed using flexAnalysis 3.4.76 (Bruker). Mass lists were generated from spectra evaluating peaks using "centroid" function, and allowing a maximum of 300 peaks to be detected. To colorize and label spectra, Affinity Designer was used. ChemDraw Professional was used to generate possible oligomer structures and isotope patterns. Macro-enabled Excel was used with an open-source macro called "MakeUpANumber", and compared with the built-in solver in Excel to find possible combinations of m/z distances in the acquired MALDI-TOF mass lists to speed up analysis.

FEDHP, ENZDHP, ACFEDHP and MWLS were dissolved in acetonitrile:MilliQ+0.1%TFA mixtures 30:70 (TA30 + 0.1% TFA) at 1 mg ml$^{-1}$ concentration. Due to MWLSA not being soluble in TA30 + 0.1% TFA, it was instead solubilized in TA50 + 0.1% TFA at 1 mg ml$^{-1}$ concentration. Matrix consisting of 2,5-DHB was prepared at 20 mg ml$^{-1}$ in TA30 + 0.1% TFA. Sample solution and matrix solution was mixed at 1:1 ratio and vortexed. The final sample-matrix mixture was deposited (0.5 μL) onto a Bruker Anchorchip MTP 384 MALDI-plate and left to air-dry. Evaporation of solvent and crystallization was achieved in less than 5 minutes for all samples. 2,5-DHB matrix solution was prepared the same day as analysis was performed and always used fresh.

## MALDI-LIFT-TOF/TOF (MS²)

The same MWLSA prepared in TA50 + 0.1% TFA for MALDI-TOF was used for MALDI-LIFT-TOF/TOF. Saturated HCCA matrix in TA50 + 0.1% TFA was prepared by addition of HCCA to TA50 + 0.1% TFA until a small pellet was visible in the bottom of the Eppendorf tube. The matrix-containing solution was centrifuged briefly to spin down non-solubilized particles. The supernatant of HCCA matrix solution and MWLSA were mixed at 1:1 ratio and vortexed. The final sample-matrix mixture was deposited (0.5 μL) onto a Bruker AnchorChip MTP 384 MALDI-plate and left to air-dry. Evaporation of solvent and crystallization was achieved in less than 10 minutes for all sample spots. Parent spectra were collected with laser beam attenuation of 43, laser beam focus of 70, and a reflector detector voltage of 2.107 kV. Fragment spectra were collected with laser beam attenuation of 20.2, laser beam focus of 70, and a reflector detector voltage of 2.212 kV. LIFT voltage 1 was 19 kV, LIFT voltage 2 was 3.7 kV. Reflector voltage 1 was 29.5 kV, and reflector voltage 2 was 14 kV. Parent and fragment spectra were saved separately, and not merged. Parent spectra were collected using 8.000 laser shots, and fragment spectra were collected using 20.000 laser shots. 1038.709 m/z and 1080.907 m/z parent masses were chosen after peak evaluation in MALDI-TOF MS mode using HCCA matrix, and the peak in the MS² parent spectrum closest to the chosen mass was selected for fragmentation.

Mass-lists obtained from MALDI-LIFT-TOF/TOF are maximized to 200 identified peaks. All of these peaks have S:N ≥ 3. Peaks from MS² were evaluated using SNAP function in the flexAnalysis software.

## Means and limitations of data analysis

Constructing model dimers and calculating their expected m/z as sodium adducts gives the starting anchor for a lignin oligomer. Following this, to find possible lignin oligomer sequences, these sequence m/z will simply be a linear combination of the starting sodium dimer adduct m/z, and the m/z increments representative of each adding bond. These linear combinations can be found by manually analyzing every peak in the MALDI-TOF MS spectra and marking the distances. This is a tedious approach, and the process can be sped up by using computational tools that allow for larger scale data analysis. In this project, the approach was kept simple by utilizing an open-source macro add-in function for Excel that allows for finding linear combinations of a given sum. With this function, the user must by themselves choose what values and how many of each value the function is allowed to utilize to try to construct the given sum. Here, the 2D-HSQC NMR inter-unit analysis is used as a guideline for chosen increments to include in the analysis.

The criteria for finding linkage-sequences are as follows:

- For pure dimers, [β-O-4' + Na]$^+$, [β-5' + Na]$^+$, and [β-β' + Na]$^+$ m/z were used as anchors for finding oligomer sequences with unaltered chemical constitution.
- For unassigned low m/z oligomers in MWLSA, 520 m/z and 758 m/z were added as additional anchors for finding oligomer sequences since these were distinct peaks which sequences could be derived from. In the LILIPOPS structure map, only 520 derived progressions are showcased but similar progression patterns could be found from the 758 m/z adduct.
- Mass-lists obtained from MALDI-TOF MS software were maximized to 300 identified peaks. All of these peaks had S:N ≥ 4. However, only the most intense peaks which could be correlated to the addition of a monomer from either low m/z anchor adduct in the spectra were evaluated. Peaks from MS were identified with a centroid function in the flexAnalysis software. Mass-lists obtained from MALDI-LIFT-TOF/TOF MS² were maximized to 200 identified peaks. All of these peaks had S:N ≥ 3. Peaks from MS² were evaluated using SNAP function in the flexAnalysis software.
- For a peak to be determined as a true identifiable oligomer, it must follow from an already determined peak that belongs to an already determined oligomer. That is, even though higher m/z peaks can be identified by the function as a linear combination of given mass increments, these are excluded if there is not a previous peak cluster it can be derived from, with a m/z distance representative of a specific known bond formation. This limits the number of identified oligomers, and it is recognized by the authors that if there are certain oligomers containing an inter-unit linkage sequence with fast polymerization kinetics that only are present in high molecular weight, then these will be lost in the identification. However, these limitations have been imposed to identify the inter-unit linkage sequences in a methodological way instead of blindly guessing structures based on m/z.
- For all found m/z through the algorithm, peaks were investigated manually to confirm that the m/z was in fact correlated to a true peak in the spectra following the established framework for linkage pattern identification described earlier. Hence, the excel-macro is not necessary for analysis but does contribute with efficiency in handling a large data set.

## Data availability

All data that support the findings of this study are available from the corresponding author upon reasonable request. Files for NMR can be found in Supplementary Data 1, and files for MALDI-TOF MS used for LPM creation can be found in Supplementary Data 2.

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

## Acknowledgements
The authors acknowledge funding from the Knut and Alice Wallenberg Foundation (KAW) through the Wallenberg Wood Science Center. The Swedish NMR centre at the University of Gothenburg is acknowledged for support.

## Author contributions
F.L: Investigation, methodology, data acquisition and analysis, and writing original draft. M.L: Conceptualization, supervision, funding acquisition, writing review, and editing. Å.E: Conceptualization, supervision, writing review, and editing.

## Funding

## Competing interests
M.L. declares being co-founder of a company, Ligolin AB, a new start-up that will valorize lignin materials.
