## [Transparent Peer Review file · Communications Chemistry]

Advancing Lignin Analytics via Elucidation of Linkage Progressions in Lignin Populations

Corresponding Author: Professor Martin Lawoko

Version 0:

Reviewer comments:

Reviewer #1

(Remarks to the Author)

This manuscript has conducted a relatively detailed analysis on the structural characterization of lignin, combining NMR, MALDI-TOF and acetylation derivatization strategies to try to establish a "structural progression map" of the connection mode. The combination of methods is of practical significance, the data is detailed, and the experimental design is rigorous. However, overall, the scientific innovation and theoretical contribution of the article have not been fully reflected. It is recommended to make in-depth revisions and reinforcements in the following aspects:

1. The concept of "Linkage Progression Map" proposed in the article is still descriptive, and its theoretical significance for the structural analysis of lignin should be further demonstrated, as well as its essential difference from the traditional "average structure" characterization. It is recommended to add a more systematic mechanism explanation or mathematical model to enhance the originality of the analysis method.
2. Deepen the depth of structural verification. There is still a lack of direct spectral evidence or synthetic control support for the proposed key structures (such as α -O-4', α -O- γ ', DBDO, etc.). It is recommended to introduce advanced 2D NMR techniques such as HMBC, TOCSY or NOESY for verification to enhance the credibility of structural assignment. For the hypothetical structure of the source of the "520 m/z" peak, synthetic standards can be considered for comparison or further MS³ verification.
3. Supplement the discussion on the applicability and limitations of the method. The current method has limited recognition ability for polymers or highly branched structures. It is recommended to supplement the applicable boundaries of this method in different molecular weight ranges and connection modes. Whether the construction of the "LILIPOPS" map can be extended to other lignin types (such as softwood/hardwood, technical lignin, etc.), it is recommended to provide a discussion or supplementary experiments.

Reviewer #2

(Remarks to the Author)

The manuscript by Lawoko and coworkers, entitled "Advancing Lignin Analytics via Elucidation of Linkage Progression in Lignin Populations", presents a novel and facile method for advancing lignin structural analysis. By combining NMR and MALDI-TOF MS, the authors demonstrate that detailed structures of low-molecular-weight molecules in milled wood lignin and synthetic lignin oligomers can be effectively distinguished and determined. The manuscript is scientifically robust and supported by extensive data, making it suitable for publication in the Communications Chemistry. However, several key points must be addressed prior to final acceptance:

Comments:

- 1) A more detailed description of the developed methods for lignin structural analysis should be included in the Introduction to provide sufficient context for readers.
- 2) NMR spectroscopy can yield valuable semi-quantitative information regarding the major inter-unit linkages in lignin. Could the MALDI-TOF MS technique also provide relative quantitative insights, and if so, how do these compare with NMR-based results?
- 3) Spruce lignin is composed primarily of guaiacyl (G) units, which makes it a suitable substrate for MALDI-TOF MS analysis. In contrast, hardwood and grass lignins exhibit greater structural diversity and complexity. The authors should clarify whether MALDI-TOF MS is equally applicable to these types of lignin, and if not, what limitations may arise.
- 4) The manuscript should be carefully revised to correct errors and inconsistencies. For instance, there is a discrepancy regarding labels A, B, and C in Figure 1 between the legend and the description (page 4, line 101). Additionally, some

references are missing (e.g., page 7 and line 121).

Version 1:

Reviewer comments:

Reviewer #1

(Remarks to the Author)

This manuscript proposes a process combining chemical derivatization with multimodal elucidation using mass spectrometry/NMR to infer the bonding patterns and sequence progression of oligomers in complex mixtures. The authors claim that this method can reconstruct the growth units and linkage patterns of several representative oligomers in both model systems and real samples. Overall, the approach is clear, with a high degree of cross-technical integration, and has potential methodological value and cross-scenario approximation. However, several core conclusions currently rely on indirect evidence or heuristics, lacking disambiguation verification, statistical robustness, and reproducibility.

1. Currently, several "first" structural assignments are primarily supported by mass increments and limited MS² fragment comparisons, which are insufficient to draw exclusive conclusions between isotopic linkages, branching, and terminal differences. Selective labeling/isotope derivatization (e.g., Ac₂O-d₆ counting hydroxyl groups) or selective chemical fragmentation experiments should be introduced for one or two representative sequences to directly verify chain orientation and attachment sites. A set of candidate structures, screening rules, and a decision tree for the final identified pathway should be provided, with the number of candidates and confidence level for each assignment reported.
2. Insufficient cross-technical constraints. Currently, NMR provides limited evidence for key long-range correlations (e.g., HMBC/HSQC-TOCSY evidence that can distinguish α -O-4 from α -O- γ). Please supplement with directed two-dimensional spectroscopy (HMBC delay optimization, HSQC-TOCSY) of the target chemical site; and employ fractionation (e.g., SEC) combined with NMR to enhance signal-to-noise and reduce peak overlap.
3. If the paper mentions structural assignments for "stable radical" or photoinduced fragmentation pathways, please provide EPR (room/cold temperature) or spin trapping controls. If this is not possible, the discussion should be downgraded to a discussion of possibilities and definitive statements should be removed.
4. Currently, validation focuses on single-prepared/processed samples. Please include at least one control sample with a different process (e.g., solvent-based/stabilized sample) to compare the consistency of key mass increments and fragmentation paths to rule out artifacts induced by sample processing.
5. Current descriptions primarily rely on qualitative terms such as "more abundant/more common." Recommendations: Provide at least $n \geq 3$ technical replicates $\times \geq 3$ locations, and provide boxplots of peak intensity relative standard deviations (RSDs) and peak position errors. For each step in the LPM/sequence plot, provide scoring or penalty rules and false positive rate (FDR) estimates (using leave-one-out validation or randomized peak controls).
6. The methodological contribution of the article should be high, ensuring reusability: Include the raw data (mzML/NMR FID), peak table (including m/z, error, S/N), and annotation script/macro with parameters, along with the article or in a supplementary publication. Provide a minimal runnable example (example dataset + one-click script) with instructions.

Reviewer #2

(Remarks to the Author)

The authors have adequately addressed all my previous concerns. I recommend acceptance.

Version 2:

Reviewer comments:

Reviewer #1

(Remarks to the Author)

The manuscript has been well-received in response to the reviewers' comments, and its quality has significantly improved. Accept after Minor revisions are recommended.

1. The manuscript already cites prior lignomics studies using MALDI-TOF and tandem MS, but the specific conceptual advance of the present work (acetylation-guided linkage progression maps for intact oligomers, applied to MWL) could be emphasized more explicitly in the Introduction and early part of the Results. Please add 2–3 sentences that clearly state how the proposed workflow goes beyond earlier studies in terms of resolving linkage progressions and assigning concrete oligomer sequences, rather than only fragmentation motifs.
2. The manuscript is generally well written, but there are scattered grammatical and stylistic issues that should be corrected at the copy-editing level. For example, "additional studied on two types of synthetic oligomeric lignin mixtures" could be revised to "additional studies were conducted on two types...", and sentences such as "the identification and determination of relative content of inter-unit linkages was determined..." can be streamlined to avoid repetition. A careful read by a native speaker or professional editing service to harmonize tense, voice, and technical phrasing would improve readability.
- 3.

2025-10-01

Dear Reviewers

We thank you for the valuable comments on our manuscript entitled “Advancing Lignin Analytics via Elucidation of Linkage Progressions in Lignin Populations “.

The comments are much appreciated by all authors. We have now addressed all the concerns and hope you will find it sufficient. Below is a point-to-point response, that is submitted together with the revised manuscript.

Thank you!

Kind Regards,

Martin Lawoko
Professor
KTH, Royal Institute of Technology

Point to point response

Red text = our response/comments

Blue text = What has been added to the paper (in response to the respective reviewer comment)

Reviewers' comments:

Reviewer #1 (Remarks to the Author):

This manuscript has conducted a relatively detailed analysis on the structural characterization of lignin, combining NMR, MALDI-TOF and acetylation derivatization strategies to try to establish a "structural progression map" of the connection mode. The combination of methods is of practical significance, the data is detailed, and the experimental design is rigorous.

Thank you for the recognition. We are humbled.

However, overall, the scientific innovation and theoretical contribution of the article have not been fully reflected. It is recommended to make in-depth revisions and reinforcements in the following aspects:

1. The concept of "Linkage Progression Map" proposed in the article is still descriptive, and its theoretical significance for the structural analysis of lignin should be further demonstrated, as well as its essential difference from the traditional "average structure" characterization. It is recommended to add a more systematic mechanism explanation or mathematical model to enhance the originality of the analysis method.

To explain the linkage progression map (LPM) concept, the following text has been added at the end of the introduction:

“To give a comprehensive overview of identified oligomers in the sample, we propose “*linkage progressions maps*” (LPMs) as a conceptualization. In an LPM, the mass-over-charge of each identified adduct is given as well as the inter-unit-linkage progression necessary to reach that exact mass. The LPM principle is based on the knowledge that lignin polymerization occurs by radical coupling and mainly by end-wise addition of one monomer at a time^{1,2}. Secondly, lignin polymerization occurs continually as long as the supply of monomers proceeds. Hence, at any given time, there will be oligomer populations that are differentiated by one monomer unit. Thus, if each linkage type can be made to contribute with a specific m/z increment when added to the growing oligomer, it becomes easy to follow the precise inter-unit linkage progression. We show herein that the fingerprinting of linkage type by m/z increment is achieved by acetylation of the sample prior to MALDI-TOF analysis. For full disclosure, there is an extensive amount of adduct peaks in the MALDI-TOF spectra of acetylated Spruce. Evaluating every single peak manually would be tedious work. We therefore showcase an LPM containing a series of oligomers stemming from one chosen starting adduct. Using this approach, we unravel for the first time in lignin analytics, the exact structures of individual oligomers in a heterogenous sample.”

Furthermore, to clarify the difference between our work and the state of the art, the figure below, together with the accompanying text “A general schematic for the mechanism can be seen in Figure 1, where the workflow traditionally used within lignin research to determine an average sample

structure is contrasted to the methodology established in this work.” has been introduced in the beginning of the results and discussions part. It also illustrates the model used to derive an LPM

Figure 1. Schematic for methodological framework developed within this project. A comparison to already established lignin literature where 2D-HSQC NMR is used as a primary analytical technique to provide an average structural motif. While powerful, traditional 2D-HSQC NMR can only give an estimate of all inter-unit linkages in a sample but not tie these to specific oligomeric structures. With the proposed workflow in this work, it is possible to reconstruct oligomers with their exact inter-unit linkage progressions tied to their exact molecular weights.

We have additionally provided a simple mathematical model under “**Linkage progression in Spruce MWL oligomer populations**” to explain the inter-unit linkage progressions and their corresponding masses in the LPM derived from adduct 520 m/z:

“For this specific and simplified analysis, a mathematical model to explain found adducts can thus be constructed as: $520 + 280x + 220y + 178z$ (where $x, y, z \geq 0, 1, 2, 3 \dots$). To reiterate, the 280 m/z signifies the β -O-4’ linkage, 220 m/z is either 4-O-5’ or β -5, and 178 m/z is the β - β ’ linkage. Note here that if there is a β - β ’ increment, the 220 residing prior in the LPM must be a 4-O-5’ due to the directionality of the chain.”

Finally, the underlying discerning mechanism for the inter-units has been added to the bottom of Figure 4 as shown below:

Unmodified and acetylated structures

$\Delta m/z$ between polymerization peaks

Unmodified and acetylated structures		$\Delta m/z$ between polymerization peaks	
Unmodified	Acetylated	Unmodified	Acetylated
1a. 381 Da 	1b. 507 Da 	+178 m/z	+220 m/z
2a. 381 Da 	2b. 465 Da 	+178 m/z	+178 m/z
3a. 399 Da 	3b. 567 Da 	+196 m/z	+280 m/z
4a. 	4b. 	+358 m/z *	+442 m/z *
5a. 	5b. 	+178 m/z	+220 m/z
6a. 	6b. 	+178 m/z **	+262 m/z **
7a. 	7b. 	+178 m/z	+178 m/z

Mechanism for discernment

Unmodified sample

Acetylated sample

Lignin Acetylated lignin

Acetylating lignin discern lignin oligomers dependent on number of OH groups. Using NMR as a guide provides further elucidation of the structure.

Figure 4. Dimer mass change and increment change for most common interunit linkages through acetylation. 1a, 6-5' (381 m/z). 2a, 6-6' (381 m/z). 3a, 6-O-4' (399 m/z). 4a, α -O-4'. 5a, 4-O-5'. 6a, 5-5'. 7a, DBDO. 1b, acetylated 6-5' (507 m/z). 2b, acetylated 6-6' (465 m/z). 3b, acetylated 6-O-4' (567 m/z). 4b, acetylated α -O-4'. 5b, acetylated 4-O-5'. 6b, acetylated 5-5'. 7b, acetylated DBDO. *Suggestion for α -o-4' (or α -O- γ ' as alternative) being an observed 442 m/z increment in acetylated sample. **5-5' is suggested by literature to occur as a dimerization reaction, and not as an addition of one single monomer to a growing chain. This increment is not included in the analysis, but the overall contribution of a theoretical monomeric unit to the observed 5-5' m/z is showcased to illustrate the mass contribution this unit has in an oligomer. A brief illustration of the acetylation mechanism employed for discernment is given at the bottom of the figure.

2. Deepen the depth of structural verification. There is still a lack of direct spectral evidence or synthetic control support for the proposed key structures (such as α -O-4', α -O- γ ', DBDO, etc.).

Alkyl-O-alkyl structures e.g. α -O- γ are generally in very low concentrations making these hard to verify by HMBC. However, recent model compound studies by Potthast et al. in "Exploring Alkyl-O-Alkyl Ether Structures in Softwood Milled Wood Lignins" J.Agric.FoodChem.2022,71,580–591, clearly verify overlap with the hypothesized regions for these structures in HSQC of MWL. This reference has been added to the discussion.

Just like the common inter-unit linkages, HSQC signals of DBDO are well established by model several NMR studies including HMBC, and are not disputable.

It is recommended to introduce advanced 2D NMR techniques such as HMBC, TOCSY or NOESY for verification to enhance the credibility of structural assignment. For the hypothetical structure of the source of the "520 m/z" peak, synthetic standards can be considered for comparison or further MS³ verification.

In the section titled "**Support for ketones and aldehydes derived by HMBC and MALDI-TOF MS²**" we have now added HMBC results to verify the altered structures. This also provides additional support for the MS² already discussed in the original version. Figure 7 has been added and discussed as shown below. The full HMBC spectrum is in the supporting information (ESI Figure 16).

Figure 7. Deciphering the atom connectivity in altered structures through combined HMBC (400 MHz) and HSQC (900 MHz) NMR. Top left (A): Complete HMBC NMR demonstrating atom connectivity by through bond correlations. Purple coded: the aldehyde proton connectivity to C_β, C_α in allylic structures, typically appearing in the region ¹³C: 125.0-130.0 ppm, and C₁ aromatic typically around ¹³C: 128.0-130.0 ppm. Blue coded: the aromatic proton attached to C₂ correlates with aromatic carbons C₁, typically at ¹³C: 128.0 ppm, C₃ and C₄, typically at ¹³C: 145.0-152.0 ppm, and C_α in ketone, typically at ¹³C: 190.0-210.0 ppm. Pink coded: previously analyzed Hibberts ketone as reported from model studies by Miles-Barett et al³⁴ is also detected. Top Right (A*): Zoom in of the aromatic, allylic and aldehyde regions of the HMBC. Signals of specific interest are marked and color-coded. Bottom left (B): 2D-HSQC NMR with noted signals relevant for HMBC analysis. Bottom right: Classification of altered structures detected in this work. Description of used symbols and previously analyzed ketones, also detected in this work.

The altered structure was further investigated by HMBC. In the interest of elucidating the altered structure only relevant sections of both HMBC and HSQC spectra are shown in Figure 7. More specifically the aromatic, allylic and carbonyl regions of the HSQC and HMBC where analyzed. From the HMBC, clear correlations between aromatic protons (7,3-7.5 ppm) with several carbons including the carbonyl in α -ketone structures, (indicated in the bottom right structure) are visible. Furthermore, correlations of aldehydic protons (9.5-10 ppm) with allylic carbons (also indicated in the bottom right structure) are observed. The lower aromatic rings on both structures were more difficult to verify by HMBC.

Finally, we unfortunately do not have MS³ possibilities in our MALDI system but recognize the added value such analysis would have provided to the study.

3. Supplement the discussion on the applicability and limitations of the method. The current method has limited recognition ability for polymers or highly branched structures. It is recommended to supplement the applicable boundaries of this method in different molecular weight ranges and connection modes.

The following text has been added in the final section just before the conclusions:

Limitations and applicability

Certain limitations of MALDI-TOF MSⁿ should be considered. Most notably, there is discrimination towards higher molecular weight oligomers in heterogenous samples which results in the molecular weight distribution being skewed towards a lower average^{40,41}. Matrix compatibility, sample preparation, and sufficient crystallization are crucial steps for MALDI-TOF MSⁿ analysis which influence the recorded spectra⁴². Therefore, it should be duly noted that in this publication the work exclusively focuses on the low molecular weight oligomers present in MWL from Spruce. Certain lignin oligomers containing 5-5'/DBDO inter-unit linkage types, are the product of dimerization reactions and will not be easily discerned with the currently developed method of analysis. The established framework is primarily suited for oligomers stemming from end-wise polymerization, as the increments between polymerization clusters are considered. Moreover, MALDI-TOF MSⁿ is primarily qualitative and not quantitative. While the exact structures of low molecular weight oligomers residing in the sample are identified, it is not possible to tell in which relative ratio these are present. Therefore, semi-quantitative 2D-HSQC NMR is still necessary for inter-unit linkage quantitation.

Whether the construction of the "LILIPOPS" map can be extended to other lignin types (such as softwood/hardwood, technical lignin, etc.), it is recommended to provide a discussion or supplementary experiments.

The following text has been added under "Limitations and applicability":

Concerning applicability, the general concept this methodology is based upon must be considered. In its essence, the acetylation simply reveals the number of hydroxyl groups present in each lignin inter-unit linkage, making it possible to discern molecular mass differences via MS prior to and after derivatization. This difference is the principle that allows for the structural unit and inter-unit linkage progression determination. Therefore, the proposed framework can and will be extended to elucidate the exact structure of low molecular weight oligomers in other type of lignins, promising broad applicability within the lignin analytical field when investigating various types of technical-, and native-like lignins.

Reviewer #2 (Remarks to the Author):

The manuscript by Lawoko and coworkers, entitled "Advancing Lignin Analytics via Elucidation of Linkage Progression in Lignin Populations", presents a novel and facile method for advancing lignin structural analysis. By combining NMR and MALDI-TOF MS, the authors demonstrate that detailed structures of low-molecular-weight molecules in milled wood lignin and synthetic lignin oligomers can be effectively distinguished and determined. The manuscript is scientifically robust and supported by extensive data, making it suitable for publication in the Communications Chemistry.

Thank you for the recognition. We are humbled.

However, several key points must be addressed prior to final acceptance:
Comments:

1) A more detailed description of the developed methods for lignin structural analysis should be included in the Introduction to provide sufficient context for readers.

We have made this addition, as also pointed out by Reviewer 1. (See our response to point 1 made by Reviewer 1).

2) NMR spectroscopy can yield valuable semi-quantitative information regarding the major inter-unit linkages in lignin. Could the MALDI-TOF MS technique also provide relative quantitative insights, and if so, how do these compare with NMR-based results?

Our understanding is that MALDI is not quantitative, but qualitative. This is due to the fact that one must calibrate the MS system towards the exact investigated molecule for accurate quantitation. The result is dependent on various factors such as matrix compatibility and the instrument settings employed during analysis. With a highly heterogeneous sample such as lignin, this is not possible at the moment. Hence, we combined it with HSQC (which is semi-quantitative). This limitation by MALDI is partly addressed in our response to Reviewer 1 (under point 3).

3) Spruce lignin is composed primarily of guaiacyl (G) units, which makes it a suitable substrate for MALDI-TOF MS analysis. In contrast, hardwood and grass lignins exhibit greater structural diversity and complexity. The authors should clarify whether MALDI-TOF MS is equally applicable to these types of lignin, and if not, what limitations may arise.

Yes indeed, we are finalizing another manuscript on the analysis of Birch lignin that will soon be submitted. We have also earlier performed prior studies on organosolv lignins, *Lawoko et al. In "Lignin Structure and Reactivity in the Organosolv Process Studies by NMR Spectroscopy, Mass Spectrometry, and Density Functional Theory" Biomacromolecules* **24**, 2314–2326 (2023). However, the acetylation protocol was not used in that case, hence lower precision in the analysis of the inter-unit progression. This is also addressed in our response to Reviewer 1, point 3.

4) The manuscript should be carefully revised to correct errors and inconsistencies. For instance, there is a discrepancy regarding labels A, B, and C in Figure 1 between the legend and the description (page 4, line 101). Additionally, some references are missing (e.g., page 7 and line 121).

We have now made these corrections.

For Reviewers

Dear Reviewers,

We thank you for the valuable comments and agree with the thoughtful suggestions to further better this manuscript. However, we also feel that part of the suggestions will require more directed and focused efforts in separate communications. We are already working on some of the issues in our laboratory. As such, we have toned down on any definitive conclusions, added discussions on limitations of the study, and provided information on follow up studies. In addition, to showcase some limitations of LPM approach, we present some preliminary analysis on technical lignin (Softwood kraft). Furthermore, and have performed some statistical analyses where applicable, and will provide raw data files. We hope this will be adequate.

The sections affected by the revisions are highlighted in the revised manuscript, as well as in the supporting information (differently color coded).

Kind regards, Martin Lawoko

Point to Point Response

Overall, the approach is clear, with a high degree of cross-technical integration, and has potential methodological value and cross-scenario approximation. However, several core conclusions currently rely on indirect evidence or heuristics, lacking disambiguation verification, statistical robustness, and reproducibility.

1. Currently, several "first" structural assignments are primarily supported by mass increments and limited MS² fragment comparisons, which are insufficient to draw exclusive conclusions between isotopic linkages, branching, and terminal differences. Selective labeling/isotope derivatization (e.g., Ac₂O-d₆ counting hydroxyl groups) or selective chemical fragmentation experiments should be introduced for one or two representative sequences to directly verify chain orientation and attachment sites. A set of candidate structures, screening rules, and a decision tree for the final identified pathway should be provided, with the number of candidates and confidence level for each assignment reported.

We agree with the assessment. However, we believe more value from it can be derived in a separate study. Until then, we have adjusted the conclusions and added discussion to the effect that further verification is required to make definitive conclusions.

2. Insufficient cross-technical constraints. Currently, NMR provides limited evidence for key long-range correlations (e.g., HMBC/HSQC-TOCSY evidence that can distinguish α -O-4 from α -O- γ). Please supplement with directed two-dimensional spectroscopy (HMBC delay optimization, HSQC-TOCSY) of the target chemical site; and employ fractionation (e.g., SEC) combined with NMR to enhance signal-to-noise and reduce peak overlap.

We agree with this assessment. As in point one above, a substantial amount of work is required and we feel this will deserve a focused manuscript where fractional/enrichment approaches are combined with both NMR approaches and spectrometry to resolve the issue of lignin

branching/pendant groups. Nevertheless, we have toned down on our discussion as relates to this issue and added a few lines on analytical strategies that may aid in resolving the issue.

3. If the paper mentions structural assignments for "stable radical" or photoinduced fragmentation pathways, please provide EPR (room/cold temperature) or spin trapping controls. If this is not possible, the discussion should be downgraded to a discussion of possibilities and definitive statements should be removed.

We have removed definitive statements related to stable radicals.

4. Currently, validation focuses on single-prepared/processed samples. Please include at least one control sample with a different process (e.g., solvent-based/stabilized sample) to compare the consistency of key mass increments and fragmentation paths to rule out artifacts induced by sample processing.

We are presently working towards demonstration of the validity of the methodology across various lignin samples. For instance, similar studies performed on Birch mill wood lignin validates the increment approach through demonstration of additional fingerprints for syringyl related increments (e.g. common increments of 310 m/z for S-based BO4 formation are found). This paper has already been written and will soon be submitted. In another project, we will study organosolv lignins produced at different conditions using the methodology to further assess the application space.

Nevertheless, we have added MALDI analysis of non-acetylated and acetylated softwood kraft lignin in the supplementary. A short discussion showcases the need for fractionation, underlining a limitation with regard to the direct analysis of crude technical lignins by this methodology.

5. Current descriptions primarily rely on qualitative terms such as "more abundant/more common." Recommendations: Provide at least $n \geq 3$ technical replicates $\times \geq 3$ locations, and provide boxplots of peak intensity relative standard deviations (RSDs) and peak position errors. For each step in the LPM/sequence plot, provide scoring or penalty rules and false positive rate (FDR) estimates (using leave-one-out validation or randomized peak controls).

6. The methodological contribution of the article should be high, ensuring reusability: Include the raw data (mzML/NMR FID), peak table (including m/z, error, S/N), and annotation script/macro with parameters, along with the article or in a supplementary publication. Provide a minimal runnable example (example dataset + one-click script) with instructions.

For sampling in MALDI-TOF MS, the spectrum is a sum of 4000 laser shots taken from 4 different parts of the sample, i.e., every spectrum is an average of distinct sample locations (>3). We ran a total of 6 replicates, and have done the statistical analysis on all 6. 3 representative spectra from both inner and outer crystallization site are provided and we have added statistical analyses that relates to points 5 and 6; mean, SD and RSD for M/z and S/N. A table with short description has been added to the supplementary information. We were unable to create scripts due to lack of a licensed software (flexAnalysis) available for non-paying users. However, we are ready to submit the raw data for both NMR and MALDI-TOF MS if we are supplied with a submission guide for how to deposit these data sets.

Point to point Response

Dear Editor and Reviewers,

We thank you once again for your comments. We are grateful for the contributions you have made to improve the quality of this manuscript. The corrections have now been made and we hope they have addressed the remaining concerns. All changes in the revised manuscript are marked in green. The manuscript has also been grammatically reviewed.

Kind Regards, Martin Lawoko

Revision 2-Point to Point-Reviewer response

The manuscript has been well-received in response to the reviewers' comments, and its quality has significantly improved. Accept after Minor revisions are recommended.

1.The manuscript already cites prior lignomics studies using MALDI-TOF and tandem MS, but the specific conceptual advance of the present work (acetylation-guided linkage progression maps for intact oligomers, applied to MWL) could be emphasized more explicitly in the Introduction and early part of the Results. Please add 2–3 sentences that clearly state how the proposed workflow goes beyond earlier studies in terms of resolving linkage progressions and assigning concrete oligomer sequences, rather than only fragmentation motifs.

We have now added a few lines to emphasize the conceptual advance.

2.The manuscript is generally well written, but there are scattered grammatical and stylistic issues that should be corrected at the copy-editing level. For example, “additional studied on two types of synthetic oligomeric lignin mixtures” could be revised to “additional studies were conducted on two types...”, and sentences such as “the identification and determination of relative content of inter-unit linkages was determined...” can be streamlined to avoid repetition. A careful read by a native speaker or professional editing service to harmonize tense, voice, and technical phrasing would improve readability

The manuscript has now been grammatically reviewed.